# Dissociating frontoparietal brain networks with neuroadaptive Bayesian optimization

Romy Lorenz [1], Ines R. Violante[2], Ricardo Pio Monti[3], Giovanni Montana[4,5], Adam Hampshire[1] & Robert Leech [6]

Understanding the unique contributions of frontoparietal networks (FPN) in cognition is challenging because they overlap spatially and are co-activated by diverse tasks. Characterizing these networks therefore involves studying their activation across many different cognitive tasks, which previously was only possible with meta-analyses. Here, we use neuroadaptive Bayesian optimization, an approach combining real-time analysis of functional neuroimaging data with machine-learning, to discover cognitive tasks that segregate ventral and dorsal FPN activity. We identify and subsequently refine two cognitive tasks, Deductive Reasoning and Tower of London, which maximally dissociate the dorsal from ventral FPN. We subsequently investigate these two FPNs in the context of a wider range of FPNs and demonstrate the importance of studying the whole activity profile across tasks to uniquely differentiate any FPN. Our findings deviate from previous meta-analyses and hypothesized functional labels for these FPNs. Taken together the results form the starting point for a neurobiologically-derived cognitive taxonomy.

[1] Department of Medicine, Computational, Cognitive and Clinical Neuroscience Laboratory (C3NL), Imperial College London, London W12 0NN, UK. [2] School of Psychology, Faculty of Health and Medical Sciences, University of Surrey, Guildford GU2 7XH, UK. [3] Gatsby Computational Neuroscience Unit, University College London, London W1T 4JG, UK. [4] Department of Mathematics, Imperial College London, London SW7 2AZ, UK. [5] Department of Biomedical Engineering, King's College London, London SE1 7EH, UK. [6] Centre for Neuroimaging Science, King's College London, London SE5 8AF, UK. These authors contributed equally: Ines R. Violante, Ricardo Pio Monti. Correspondence and requests for materials should be addressed to R.L. (email: lorenz.romy@gmail.com) or to R.L. (email: robertleech6@gmail.com)

It is well established that cognition is an emergent property of distributed networks in the brain[1], and that a set of fronto-parietal networks (FPNs) plays a particular prominent role in cognitive processes[2,3]. However, despite being the focus of intensive research efforts, the unique functional role of each FPN remains poorly understood[4,5].

The most notable reason for this failure is that it is remarkably difficult to predict which cognitive tasks will isolate a FPN based on cognitive psychology theory[6–8]. The classic taxonomy of cognitive processes was developed largely blind to the functional organization of the brain; therefore, classic cognitive tasks tend to tap complex processes that involve multiple networks[8–10]. This leads to seemingly paradoxical observations when studying the functional role of FPNs: on the one hand, FPNs are commonly co-activated during a diverse range of cognitive conditions; this is even the case when performing tasks that were originally designed to assess putatively distinct cognitive processes[11–13]. On the other hand, tasks that were originally designed to tap the same process can activate different FPNs[14]. This problem of overlapping functional profiles is exacerbated by the fact that FPNs also overlap spatially[13,15–18].

Resolving this many-to-many mapping[10] problem between cognitive tasks and brain networks is practically intractable with standard neuroimaging methodology because the cost and difficulty of data acquisition using functional magnetic resonance imaging (fMRI) necessitates testing only a small subset of all possible cognitive tasks. This is problematic, as studying only a fraction from the large task space can result in over-specified inferences about functional-anatomical mappings with a misleadingly narrow function being proposed as the definitive role of a network, concealing the broader role each network may play in cognition[5,16,19–22]. In turn, this can result in inappropriately focused theories that may bias the field and potentially fuel limited generalizability and reproducibility of neuroimaging findings[23,24]. In the context of this problem, a more holistic understanding of the functional roles of distinct FPNs is required, and this necessitates a more comprehensive approach that examines how FPN activities vary across diverse cognitive tasks.

One such approach is to use meta-analyses, synthesizing across thousands of neuroimaging findings[25,26]. While meta-analyses are powerful for tackling research questions about broad cognitive domains, they cannot extract information about fine-grained cognitive states[25]. In addition, they are prone to be affected by biases in the literature, e.g., which experiments were run, the reporting of results (e.g., file-drawer effect[27]), selected contrasts, inconsistent labeling of brain areas, and cognitive states[8,9,22], as well as variable acquisition and analysis techniques[28].

To overcome these limitations in current human brain mapping methodology, we have recently proposed a radically different approach: neuroadaptive Bayesian optimization[29]. This technique is characterized by a closed-loop search through a large task space, with fMRI data analyzed in real-time and the next task to be selected based on the real-time results. This approach allows building upon pre-existing knowledge about the functional organization of the brain by using meta-analyses as a starting point to define a prior model of how cognitive tasks map to, for example, FPN activity[23]. It then continuously validates and refines that model in an iterative learning cycle on a subject-by-subject basis, whereby predictions are generated at a given iteration and experimentally tested in the next iteration. This produces a robustly validated model across a potentially high-dimensionality space while simultaneously identifying task conditions that produce the optimal outcome[30,31].

Here, we apply this approach for the first time to study the functional specialization of a dorsal FPN (dFPN) and ventral FPN (vFPN), both of which are core parts of the multiple demand system and known to co-activate across diverse cognitive conditions[11].

We achieve this by first identifying a large pool of cognitive tasks (Fig. 1a) that recruit these two networks based on a previous meta-analysis[26]. This is followed by three real-time optimization experiments addressing different research questions of varying precision and complexity. In Experiment 1, we seek to discover the cognitive tasks that maximally dissociate the dFPN from the vFPN by searching across the meta-analytic-derived task space in real-time (Fig. 1a). Contrary to the previous meta-analysis[26], we find the Tower of London and Deductive Reasoning tasks best segregate the dFPN from the vFPN. In Experiment 2, we further maximize the dissociation between these two networks by fine-tuning specific design parameters of the two tasks identified in Experiment 1 (Fig. 1b). We demonstrate that increasingly complex relational integration and multi-step planning modulate the dFPN from vFPN dissociation in the Deductive Reasoning and Tower of London task, respectively. In Experiment 3, we investigate the unique functional profile for these two FPNs by going beyond a two-FPN to more challenging multiple-FPN dissociation. Our results suggest that for both the dFPN and vFPN: (i) the meta-analysis only partially predicts the set of optimal tasks identified by neuroadaptive Bayesian optimization; (ii) there is not a single optimal task but instead it is the functional profile across the many tasks (the whole task space) that is unique to each network—indicative of a complex many-to-many mapping between cognitive tasks and FPNs; and (iii) the set of tasks identified only partially correspond to previous functional descriptions made in the literature and do not necessarily share a prima facie intuitive underlying cognitive label or process.

## Results

**Closed-loop neuroadaptive Bayesian optimization**. Prior to the real-time experimentation, a 2D-task space was designed with each dimension corresponding to the probability of 16 different cognitive tasks recruiting the dFPN and vFPN (Fig. 1a) according to a previous meta-analysis[26]. Based on this meta-analysis, we predicted the Wisconsin Card Sorting and Counting/Calculation task to be optimal for maximally dissociating the dFPN from the vFPN while we would expect the Posner, Anti-Saccade and Go/No-go tasks to be best for maximally discriminating the vFPN from the dFPN. For the three real-time experiments, we employed a neuroadaptive Bayesian optimization approach as described in Lorenz et al.[29] and summarized below. The aim of Experiment 1 and Experiment 2 was to identify the cognitive tasks or task parameters, respectively, that maximally dissociated the two spatially overlapping FPNs; therefore, the target brain state that we optimized for was the difference in evoked blood-oxygen-level dependent (BOLD) signal of the dorsal over ventral FPN (dFPN > vFPN). The aim of Experiment 3 was to identify, in two separate runs, the optimal cognitive tasks when dissociating either the dFPN or the vFPN against all remaining (two additional) FPNs from the meta-analysis. We then searched again across the same task space from Experiment 1 in real-time, but this time the target brain state varied across the two runs and was the difference in BOLD signal of the dFPN or vFPN over the mean activity of three other FPNs (i.e., dFPN > mean(FPNs) or vFPN > mean(FPNs)).

Neuroadaptive Bayesian optimization can be understood as a two-stage procedure that repeats in an iterative closed-loop. The first stage is the data modeling stage, in which the algorithm uses all available observations obtained from real-time fMRI up to that point to predict the subject's brain response across the entire task space. We used Gaussian process regression as our model, due to its versatility and flexibility[32]. The resulting model ("surrogate

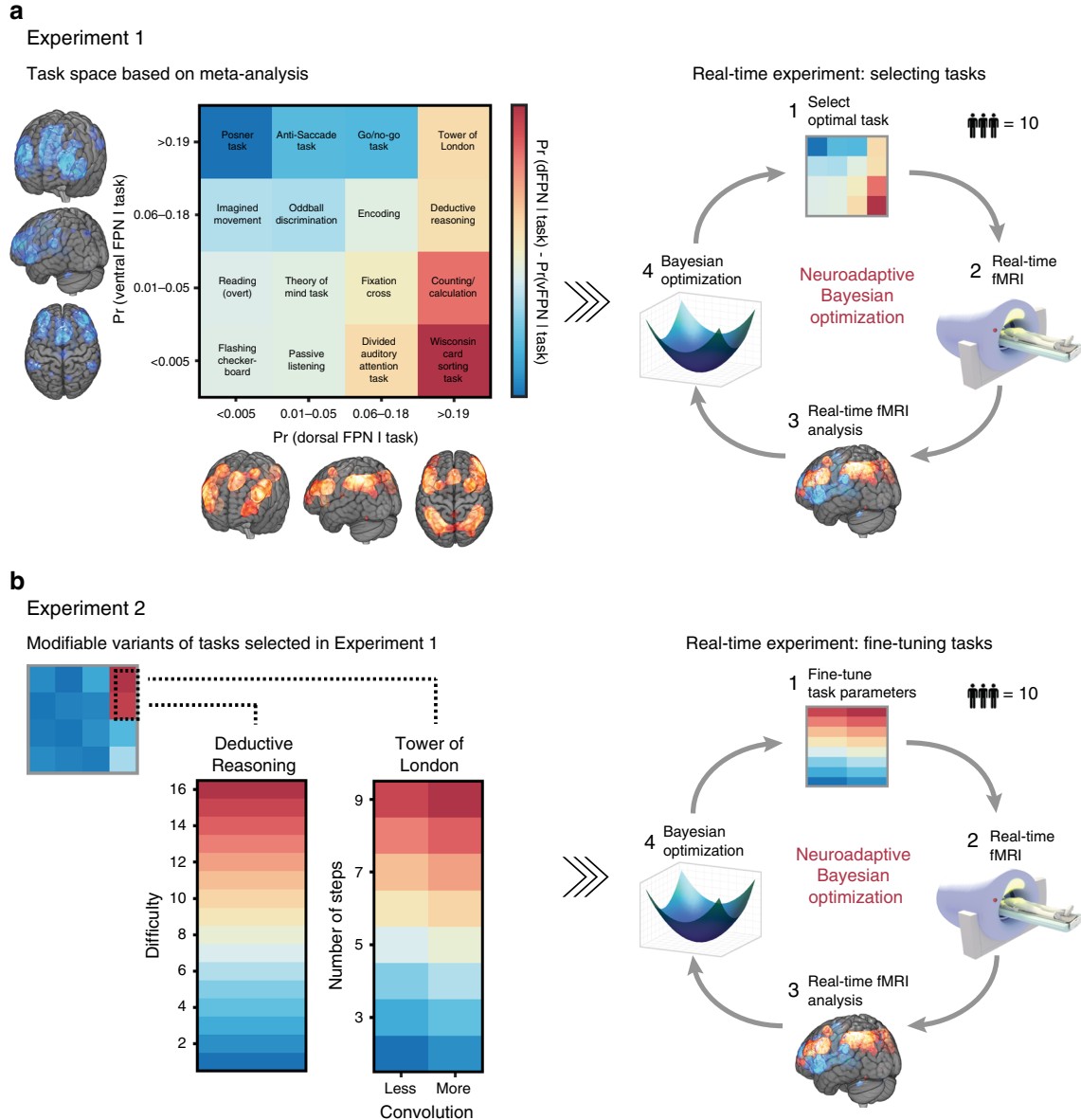

**Fig. 1** Overview of methodology. **a** In Experiment 1, a 2D-task space was designed with each dimension corresponding to the probability of 16 tasks recruiting the dFPN or vFPN according to a previous meta-analysis[26]. Color-coding indicates the hypothesized dissociation (based on this meta-analysis) between the two FPNs: red indicates tasks to be optimal for the contrast dFPN > vFPN (i.e., Wisconsin Card Sorting and Counting/Calculation tasks), while blue indicates tasks to be optimal for the reverse contrast vFPN > dFPN (i.e., Posner, Anti-Aaccade and Go/no-go tasks). This task space was then searched in real-time by the neuroadaptive Bayesian optimization to find optimal tasks that dissociate the dFPN from the vFPN. **b** In Experiment 2, task parameters of optimal tasks from Experiment 2 were fine-tuned. For the Deductive Reasoning task, the optimization algorithm searched across 1D-space with 16 × 1 difficulty levels. For the Tower of London task, the optimization algorithm searched across 2D-space with 8 (number of steps) x 2 (convolution) parameters. In Experiment 3 (not shown), the same 2D-task space as in Experiment 1 was searched through by the optimization algorithm with the aim of finding task that maximally dissociate the dFPN and vFPN from three other FPNs (for details see main text)

model") captures the algorithm's beliefs about the relationship between the task space and the subject's brain response.

The second stage is the guided search stage, in which an acquisition function is used to propose a point in the task space to sample next (i.e., the task, the subject will need to perform in the next iteration). This new observation will then be used to update the algorithm's surrogate model. The acquisition function determines the "usefulness" of every point in the task space for achieving its learning goal (i.e., finding a set of tasks that maximize the respective BOLD contrast) and directs the sampling to the most "useful" point at a given iteration; this allows for an efficient and reliable search over an exhaustive task space. The

definition of "usefulness" can vary depending on the specific acquisition function used. Here, we employed the expected improvement acquisition function, that is characterized by an automatic transition from explorative to exploitative search behavior when optimization is successful[30]. Thus, the algorithm starts with an exploration phase by obtaining samples (i.e., BOLD contrasts) across the task space; once the algorithm's uncertainty about the task space decreases (as it learns the relationship between the tasks space and the subject's brain response), it transitions into an exploitative phase, in which the acquisition function keeps sampling the predicted optimum or nearby in the task space. This resembles an inbuilt replication stage: every new

task or task condition proposed in real-time by the algorithm serves as a new test sample to validate the algorithm's predictions.

This procedure was performed for each run (Experiment 1) and each subject (Experiment 1–3) separately, i.e., the algorithm was completely blind to any data collected in the subject's previous run or any previous subjects. Therefore, from a statistical point of view, each run and each single subject in our experiments can be considered as a complete new validation test set in itself. In addition to presenting group-level results here, we provide all subject-level results in Supplementary Figs. 3, 5, 7 and 8.

**Maximally dissociating dFPN from vFPN**. In Experiment 1, the optimization algorithm successfully searched the 2D-task space (Supplementary Fig. 1a); this was also reflected in the subject-level search behavior (Supplementary Fig. 2). Across all 10 subjects (2 runs each), we observed that both the Tower of London

and the Deductive Reasoning tasks were the most frequently sampled tasks (each over 30% compared to 10% and below for other tasks) by the acquisition function (Fig. 2a). Such sampling behavior is highly reflective of identified optima as described above. Equally, when estimating a group-level Bayesian model (i.e., Gaussian process regression) based on all observations, the Tower of London and Deductive Reasoning tasks were predicted to be optimal for dissociating the two networks (Fig. 2b). This result was qualitatively highly consistent within and across subjects (Supplementary Fig. 3) and could not be explained by the particular spatial arrangement of the tasks (Supplementary Results) or a negative induced BOLD activation in the vFPN for these tasks (Supplementary Fig. 4). Post-hoc analyses, assessing the spatial similarity of the group-level predictions within different sub-regions of the dFPN and vFPN, confirmed that neuroadaptive Bayesian optimization chose tasks that selectively activated the entire dFPN, and not only sub-regions (Fig. 3a).

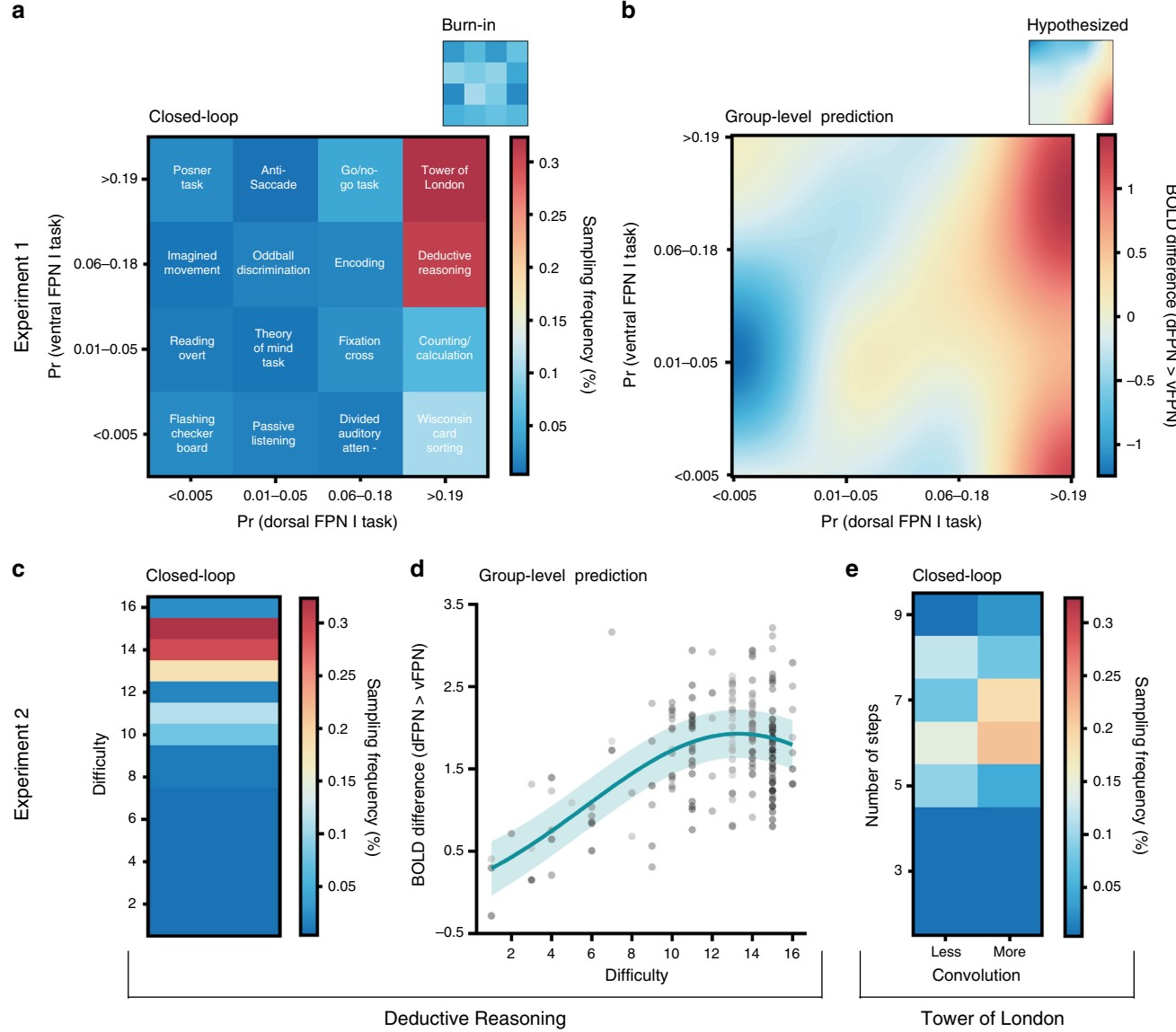

**Fig. 2** Group-level results of Experiment 1 and Experiment 2. **a** Real-time sampling behavior of neuroadaptive Bayesian optimization and **b** group-level predictions across task space identify the Tower of London and Deductive Reasoning tasks to be optimal for dissociating the dFPN from the vFPN. **c** Real-time sampling behavior of optimization algorithm and **d** group-level predictions (observations colored by subject) show difficulty levels 13–15 to be optimal for the Deductive Reasoning task. **e** Real-time sampling behavior of optimization algorithm for the Tower of London task suggests 6–7 steps to be optimal

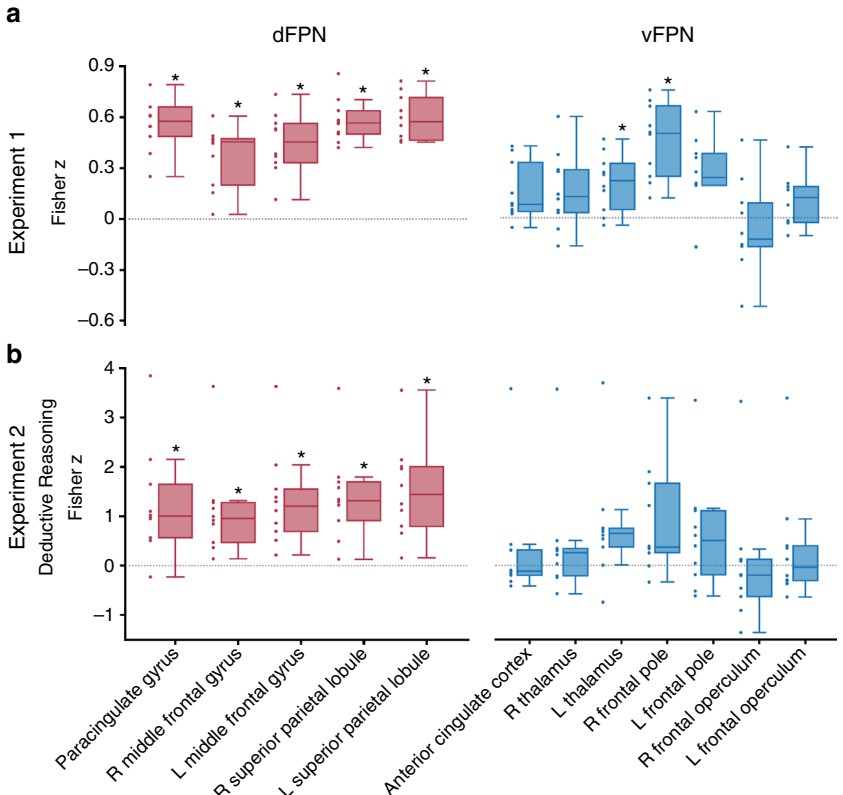

**Fig. 3** Post-hoc cluster-based analyses within dFPN and vFPN. To demonstrate that the optimization method selected tasks (Experiment 1) and task parameters (Experiment 2) that selectively targeted the entire dFPN, we performed post-hoc whole-brain back-projections (see Methods section—Post-hoc cluster-based analyses within dFPN and vFPN). Asterisks (*) indicate significant findings ($p < 0.05$) after correcting for multiple comparisons using permutation testing (critical $t(9) = 3.46$ for Experiment 1; critical $t(9) = 2.63$ for Experiment 2). **a** We find that for Experiment 1 identified tasks optimally increased the contrast with the vFPN across the entire dFPN, i.e., our obtained real-time results were not driven by single clusters within the dFPN. In comparison, most regions within the vFPN did not differ from zero; this was expected (since we contrasted beta-coefficients from each voxel's time series with the beta-coefficients of the vFPN). **b** This result was replicated for the Deductive Reasoning task in Experiment 2. This analysis was not carried out for the Tower of London task because the group-level predictions were different from the subject-level results (see Supplementary Fig. 6). On each box, the central mark is the median, the edges of the box are the 25th and 75th percentiles, the whiskers extend to the most extreme datapoints considered not to be outliers. Left to each plot are all individual observations ($n = 10$ for both experiments)

**Optimal tasks are contrary to predictions by meta-analysis**. The finding that the Tower of London and Deductive Reasoning tasks best segregated the dFPN and vFPN was inconsistent to the initial meta-analysis[26] that predicted the Wisconsin Card Sorting and Counting/Calculation tasks to be optimally suited for this purpose (Fig. 2b - small panel). This unexpected finding was statistically assessed by comparing the spatial similarity between voxel-wise Bayesian predictions and the predictions derived from the meta-analysis across the whole task space. This confirmed that the meta-analysis was significantly different from the obtained group-level results within the whole dFPN ($t(9) = 5.62$, $p < .001$, paired two-tailed $t$-test). For a more focused evaluation, the same analysis was performed for five separate clusters within the dFPN. In line with the previous result, it revealed that for each cluster within the dFPN the obtained group-level results were significantly more likely than the hypothesized predictions (Fig. 4).

**Relational integration modulates dFPN from vFPN dissociation**. In Experiment 2, the parameters of the optimal tasks from Experiment 1 were fine-tuned with an additional 10 subjects, producing a network dissociation that was even finer-grained. For the Deductive Reasoning task we successfully optimized (Supplementary Fig. 1b) across a 1D-space with 16 × 1 difficulty levels (see Methods section—Experiment Space). Across all subjects, difficulty levels 13–15 were most frequently selected by the

acquisition function during the real-time period (Fig. 2c). This result was confirmed when estimating a group-level Bayesian model (Fig. 2d) and was qualitatively highly consistent across all subjects (Supplementary Fig. 5). Linear mixed-effect analyses indicated a significant quadratic relationship between difficulty and brain activity ($\chi^2(1) = 12.07$, $p < .001$, likelihood ratio test), suggestive of plateauing performance with difficulty (Fig. 2c,d). Difficulty levels 13–15 encoded Deductive Reasoning problems involving high relational integration (see Methods section—Experiment Space). Assessing spatial similarity of the group-level predictions within separate sub-regions of the dFPN and vFPN, confirmed that these task parameters selectively activated the entire dFPN (Fig. 3b).

**Number of steps modulates dFPN from vFPN dissociation**. The Tower of London task was successfully optimized (Supplementary Fig. 1c) across an 8 (number of steps) ×2 (convolution) parameter space (see Methods—Experiment Space). Across all subjects, spatial planning problems involving 6 to 7 steps were most frequently selected by the acquisition function during the real-time period (Fig. 2e). When estimating a group-level Bayesian model for the Tower of London task, distinct predictive patterns for the two convolution stages across the experiment space were observed (Supplementary Fig. 6a); however these effects seemed to be driven by a few individuals (Supplementary

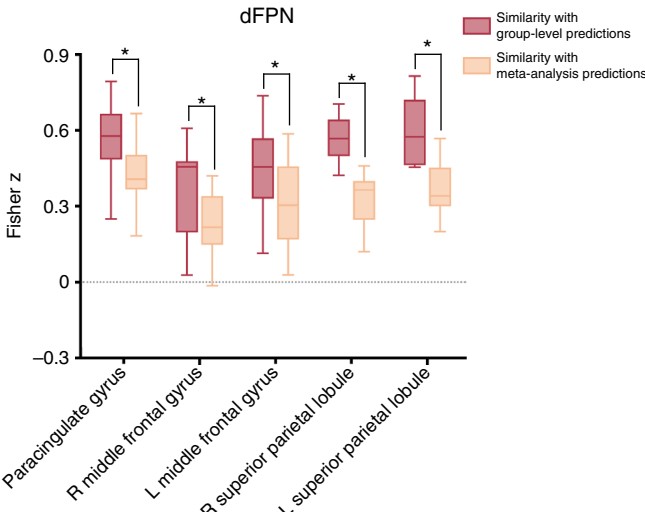

**Fig. 4** Post-hoc cluster-based comparison with meta-analysis. In order to statistically assess differences between our obtained group-level Bayesian predictions from Experiment 1 and the, according to the meta-analysis, hypothesized predictions across the whole task space, we compared two different post-hoc whole-brain back-projections (see Methods section—Post-hoc cluster-based comparison with meta-analysis). Asterisks (*) indicate significant findings ($p < 0.05$) after correcting for multiple comparisons using permutation testing (critical $t(9) = 2.81$). We found that for each cluster within the dFPN the obtained group-level results were significantly more likely than the hypothesized predictions. On each box, the central mark is the median, the edges of the box are the 25th and 75th percentiles, the whiskers extend to the most extreme datapoints considered not to be outlier

Fig. 6b). This assumption was confirmed when conducting a linear mixed-effect analysis that showed a significant quadratic effect of number of steps with brain activity ($\chi^2(2) = 76.18$, $p < 0.001$, likelihood ratio test) but no significant effect of convolution ($p = 0.92$) or interaction ($p = 0.95$). Qualitatively assessing the predicted optimal task parameters on a subject-level again indicated inter-subject reliability (Supplementary Fig. 7).

**Limitations of Experiment 1 and Experiment 2**. The results of Experiment 1 and 2 were obtained for the specific contrast of the dFPN greater than the vFPN. In this context, our approach investigated a many-to-one mapping: many different experimental task conditions were tested to optimize a single contrast in BOLD signal. There are two fundamental limitations to the functional interpretation of our results from this approach. First, the results obtained cannot be used to infer the optimal tasks for the opposite contrast: vFPN greater than the dFPN. This is because the acquisition function sampled the experiment space unevenly to maximize dFPN activity, sampling some regions in the space intensively, and leaving other areas (particularly those related to vFPN activity) under-sampled. Second, the approach falls short of conveying the unique functional role of any of these two FPNs (i.e., what processes do these networks do that other FPNs do not). This can only be addressed when properly accounting for the many-to-many-mapping that captures the idea that no one task differentiates a network from all others, but rather, each network has a unique activation profile across multiple tasks. Functional networks must therefore be defined according to how they uniquely pivot within that multivariate task space. For this reason, in Experiment 3 we studied the functional profile of the dFPN and vFPN in the context of a wider range of FPNs: an inherently more challenging aim. Accordingly,

we went back to the meta-analysis[26] and selected all remaining FPNs (in addition to the dFPN (Component 09) and vFPN (Component 08)), resulting in two additional FPNs: a left-lateralized FPN including the inferior frontal gyrus (Component 05) and a bilateral FPN distributed across the medial frontal cortex, the superior parietal cortex and frontal eye fields (Component 06). We then searched again across the task space in real-time, providing a full double dissociation of the two networks as well as information pertaining to the many-to-many-mapping between cognitive processes and brain networks.

**Maximally dissociating one FPN from multiple FPNs**. Rather than optimizing for a single contrast between two networks, in Experiment 3, we optimized one network against three others FPNs from the meta-analysis. That is, we optimized in one run for the contrast (1) dFPN > average of the remaining three FPNs. To produce an alternative dissociation, in the other run, we optimized for the contrast (2) vFPN > average of the remaining three FPNs. For both contrasts, the optimization algorithm successfully searched the 2D-task space (Supplementary Fig. 1d–e).

For the contrast dFPN > the other three FPNs, across all ten subjects, we observed that the Tower of London, the Wisconsin Card Sorting and Encoding tasks were most frequently sampled by the acquisition function (Fig. 5a). Similarly, when estimating a group-level Bayesian model (i.e., Gaussian process regression) based on all observations, the Tower of London, Wisconsin Card Sorting and Encoding tasks were predicted to be optimal for disambiguating the dFPN from the other FPNs, in addition to the Deductive Reasoning task (Fig. 5b). Overall we found a similar activation profile across the entire task space when compared to the single dissociation from the vFPN obtained before (Fig. 2b); however, when accounting also for the other two FPNs, the Wisconsin Card Sorting task exhibits similar differential activation as the Tower of London and Deductive Reasoning tasks. This is more similar to the meta-analyses (although still notably different), whereby the Wisconsin Card Sorting task was hypothesized to be most uniquely associated with this network (Fig. 5b - small panel).

For the contrast vFPN > remaining three FPNs, we found the Go/No-go, Imagined Movement, and Passive Listening tasks to be most frequently sampled across all subjects (Fig. 5c). When estimating a group-level Bayesian model, equally the Go/No-go, Imagined Movement, and Passive Listening tasks were predicted to be optimal for disambiguating the vFPN from the other three FPNs, in addition to the Reading, Encoding, and Divided Auditory Attention tasks (Fig. 5d). According to the meta-analysis, the Go/No-go task was hypothesized to be most uniquely associated with this network (Fig. 5d - small panel), while the other tasks were more unexpected.

It should be noted that for both contrasts the acquisition function sampled more broadly than in Experiment 1. This can also be observed when considering individual subjects' results (Supplementary Fig. 8), indicating less consistent optima across subjects. However, in an additional analysis, we observe that the group-level results are consistent with the individuals' results (spatial correlation coefficients between group and subject-level results in Supplementary Fig. 8). We discuss this in more detail below.

**Unique functional profiles for dFPN and vFPN**. We performed post-hoc analyses to assess the suitability of our target measure (i.e., dFPN/vFPN > average of the remaining three FPNs) for identifying the unique functional activation profile for the dFPN and vFPN. For this purpose, we pooled all available data from Experiment 1 and Experiment 3 and computed each pairwise

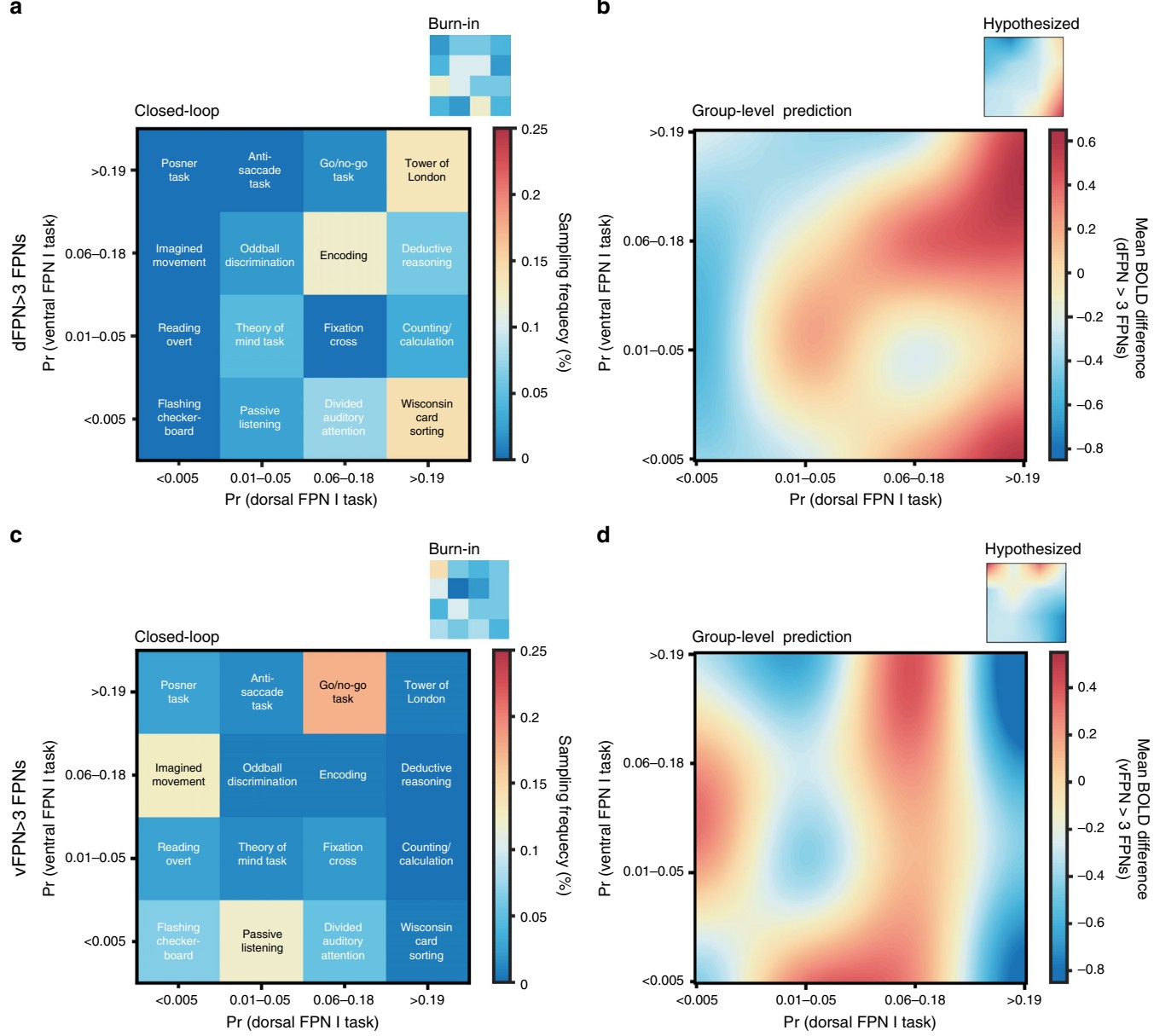

**Fig. 5** Group-level results of Experiment 3. **a** For the contrast dFPN > all other three FPNs, the Tower of London, Wisconsin Card Sorting, and Encoding tasks were most frequently sampled by neuroadaptive Bayesian optimization. **b** Group-level predictions across the task space equally show the Tower of London, Wisconsin Card Sorting and Encoding tasks to be optimal for dissociating the dFPN from the other FPNs, in addition to the Deductive Reasoning task. **c** For the contrast vFPN > all other three FPNs, the Go/No-go, Passive Listening and Imagined Movement tasks were most frequently sampled by neuroadaptive Bayesian optimization. **d** Group-level predictions across the task space equally show the Go/No-go, Passive Listening and Imagined Movement tasks to be optimal for dissociating the vFPN from the other FPNs, in addition to the Reading and Divided Auditory Attention tasks

comparison between the dFPN/vFPN and the three remaining FPNs across the entire task space (Fig. 6). For the sake of completeness, we also computed each contrast between the two FPNs added in Experiment 3 and the three remaining FPNs (Fig. 6—see Component 05 and Component 06). For all FPNs, each pairwise comparison yielded qualitatively distinct activation profiles across the task space. As detailed in the next paragraphs, the tasks or set of tasks identified as optimal for each single pairwise comparison, were all resembled in the average activation profile of the respective networks (Fig. 6 - bottom row). This demonstrated that this target measure indeed was well suited for addressing the many-to-many mapping problem by identifying a set of tasks that uniquely differentiates one FPN from functionally highly similar FPNs.

In particular, for the dFPN we found a unique activation profile (i.e., average activation profile in bottom row) featuring the Tower of London, Wisconsin Card Sorting and Deductive Reasoning tasks. Equally, these tasks or a subset of them were also present in each single pairwise comparison: for the contrast dFPN > Component 05, the Tower of London task was highly activated; for the contrast dFPN > Component 06, the Wisconsin Card Sorting, the Divided Auditory Attention, the Encoding, and the Theory of Mind tasks were highly activated, and for the contrast dFPN > vFPN, the Tower of London and Deductive Reasoning tasks were highly activated.

The average activity profile for the vFPN showed a unique activation profile featuring the Reading, Go/No-go, Imagined Movement, Divided Auditory Attention, and Passive Listening

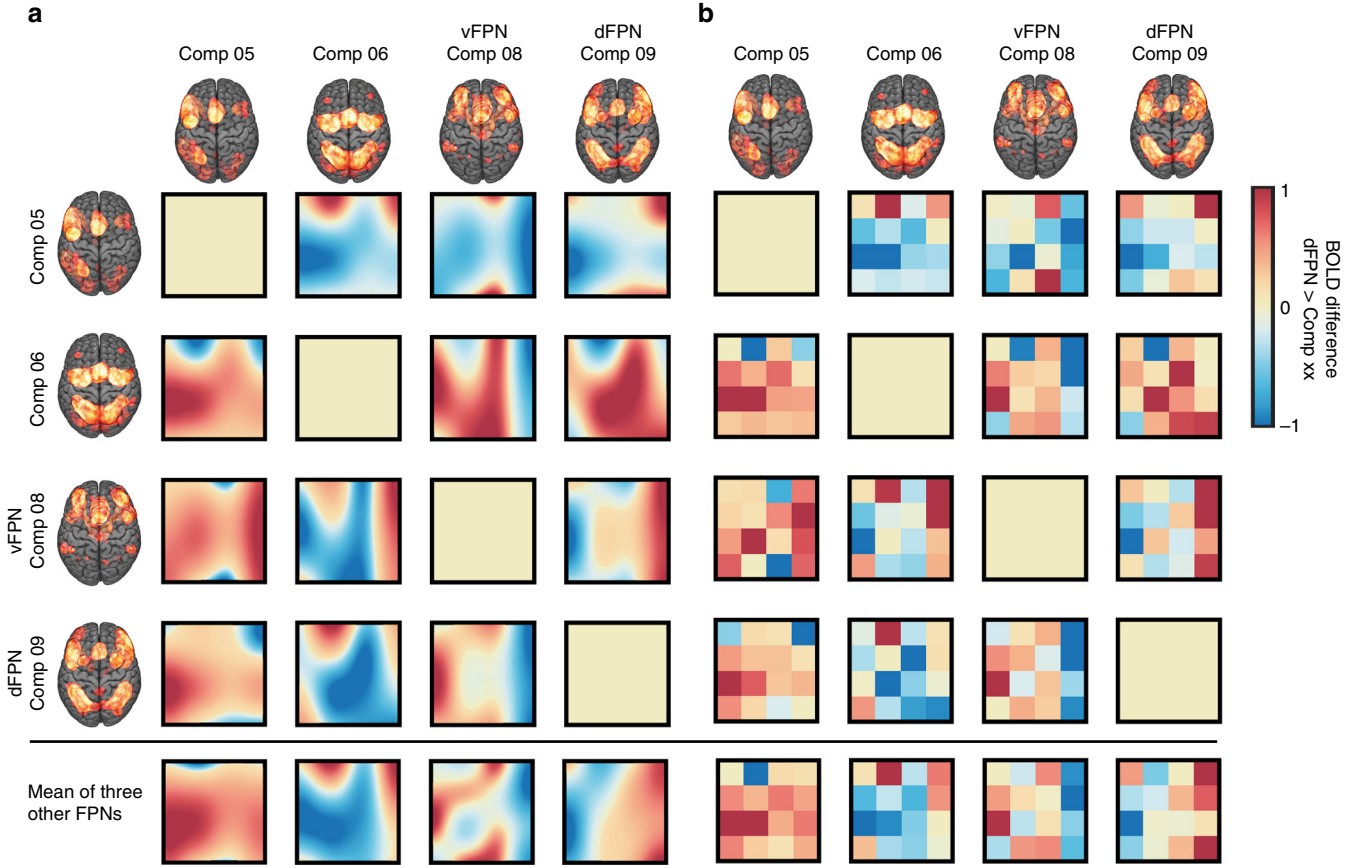

**Fig. 6** Post-hoc pairwise comparison between all four FPNs. **a** Group-level Bayesian predictions (i.e., Gaussian process regression) based on pairwise differences in BOLD between each pair of FPNs (i.e., Components (Comp) from the meta-analysis). **b** To explore if the Gaussian process regression resulted in sensible predictions, we also plotted the mean differences in BOLD between each pair of FPNs across all subjects for each cell in the task space. Data is pooled from both Experiment 1 and Experiment 3. In the bottom row, we show results for the same target measure we used in Experiment 3: difference in BOLD between each single FPN and the mean of the remaining three FPNs. Our results demonstrate the existence of a complex many-to-many mapping between cognitive tasks and FPNs as not a single task, but instead a functional profile across many tasks (i.e., the whole task space) is unique to the dFPN and vFPN

tasks. These tasks or a subset of these tasks were also present in each single pairwise comparison: for the contrast vFPN > Component 05, the Go/No-go and Divided Auditory Attention tasks were highly activated; for the contrast vFPN > Component 06, the Reading and Imagined Movement tasks were highly activated, followed by Divided Auditory Attention, Passive Listening and the Go/No-go tasks; and for the contrast vFPN > dFPN, the Reading and Imagined Movement tasks were highly activated, followed by the Go/No-go and Divided Auditory Attention tasks.

## Discussion

Over three experiments, we have used neuroadaptive Bayesian optimization to dissociate functionally similar and spatially overlapping frontoparietal brain networks. In the first experiment we discovered two tasks, the Deductive Reasoning and the Tower of London, that maximally activated the dFPN over vFPN. These results were only partially predicted by a previous meta-analysis of the existing fMRI literature; this discrepancy may reflect the biases inherent in meta-analyses. In the second experiment, by fine-tuning specific design parameters of these two tasks, we achieved a maximum dissociation between the dFPN and vFPN for complex relational integration and multi-step planning problems. While the first two experiments focused on dissociating the dFPN from the vFPN, a related but different question

concerned how these two networks relate to other FPNs identified in the meta-analysis. The third experiment revealed that it is important to consider the whole activity profile across the tasks to uniquely differentiate any single FPN from the others, suggestive of a complex many-to-many mapping between cognitive tasks and FPNs. In summary, we demonstrated that the tasks identified using neuroadaptive Bayesian optimization only partially correspond to previous functional descriptions made in the literature and do not necessarily share a prima facie intuitive underlying cognitive label or process.

In the past, a dual-network architecture of the multiple demand system has been proposed[33], describing the dFNP and vFPN as closely coupled subnetworks taking on different functional roles. In this distinction, the dFPN has been labeled as rapid, adaptive control network[34]. While there is evidence that the dFPN indeed is involved in adaptive task control[35], our results suggest that this cognitive label may not fully explain the dissociation between the dFPN and vFPN. This is because the Wisconsin Card Sorting task—a task assessing the cognitive ability to flexibly adapt to new rules, did not result in the largest dissociation between these two networks. In line with our results, Hampshire et al.[14] have previously shown that tasks associated with reasoning, such as deductive reasoning, verbal reasoning or spatial rotation more strongly engage the dFPN than the vFPN.

From a cognitive perspective, the Deductive Reasoning task (at the higher difficulty levels) involves increasingly complex

relational integration, whereas the optimal Tower of London task comprises multi-step planning. Interestingly, analogical reasoning tasks relying on visuospatial integration of rules have been associated with greater activation in the superior and inferior parietal cortex when compared to semantic analogical reasoning problems[36,37]. As both of our tasks (Deductive Reasoning and Tower of London) were of a visual rather than semantic nature, this may explain the stronger dissociation of the dFPN from the vFPN with greater demands for integrating spatial relational information. Across both tasks the dissociation between the two FPNs was less sensitive for convolution or multiple, yet non-integrated rules; such integration processes have been associated with frontopolar networks in the past[38–40]. While we found highly consistent subject-level results for the Deductive Reasoning task, we observed less consistency across subjects for the Tower of London task; possibly reflecting the increased complexity of the experiment space spanning two instead of one task dimension and/or that the effect of convolution is relatively subtle compared to the number of steps.

One possibility is that the maximum dissociation between dFPN and vFPN occurred for the two cognitive tasks that in general are more difficult than any of the other 14 tasks. Indeed, we observe an increase in the dissociation for both tasks as they become more challenging (in terms of planning steps or relational integration) before the dissociation asymptotes with increasing difficulty. However, there are other tasks that are cognitively challenging but that do not show the same dissociation (e.g., Encoding task) or show even the reverse pattern, i.e., greater activity for the vFPN than the dFPN (e.g., Divided Auditory Attention task). Similarly, if the dissociation related just to difficulty then the vFPN > dFPN contrast would be expected to be greatest for very passive tasks such as Fixation Cross. Therefore, while general task difficulty could play a part in the explanation for the dissociation for those tasks, it is unlikely to be the full explanation, and is more likely to reflect difficulty related to specific cognitive processes.

When considering the tasks that uniquely activate the dFPN compared to all other FPNs, we found a similar functional profile as identified in Experiment 1, with the Deductive Reasoning and the Tower of London being optimally suited for this one-from-many network dissociation. However, different to Experiment 1, we also identified the Wisconsin Card Sorting task to be part of the functional profile of the dFPN. The Wisconsin Card Sorting task (among others) is highly activated for the dFPN when compared to a more posterior FPN (Component 06), encompassing the frontal eye fields[41]. In contrast, the Deductive Reasoning and the Tower of London tasks both do not show any preferential activation for the dFPN in this specific pairwise comparison. In the meta-analysis[26], this posterior FPN was associated with visual spatial attention and the top task recruiting this network was the Anti-Saccade task. This was also replicated in our post-hoc analyses, where we found the functional profile of this network consisting of tasks relying on strong eye movement control, such as the Anti-Saccade, Tower of London, and Deductive Reasoning tasks. We note that the results of the Wisconsin Card Sorting task are in line with previous descriptions of the dFPN as a rapid and adaptive control network[34,35]; however, as pointed out above, our results also highlight that this functional description may be overly narrow and does not account for the Deductive Reasoning and Tower of London tasks being part of the functional profile of the dFPN.

For the vFPN, we observe a much more distributed pattern of activation across the task space; again, only partially predicted by the meta-analysis. In particular, the Go/No-go task was predicted by the meta-analysis and is broadly consistent with previous descriptions of the vFPN as involving response inhibition[26,42].

Similarly, the selection of the Divided Auditory Attention task may be attributable to other hypothesized descriptions of the vFPN like salience processing[43]. As such, this network may be responsible for detecting behaviorally relevant information, e.g., pressing a button in response to "odd" visual and auditory stimuli as is the case for our Divided Auditory Attention task (see Supplementary Methods). In line with this idea, another study has shown that pre-stimulus functional connectivity between the vFPN and auditory cortex predicted whether an auditory target was heard or missed[44]. Other tasks, such as Reading, Imagined Movements and Passive Listening are less traditionally associated with the putative cognitive processes tapping the vFPN; however we note that speech production is strongly associated with ventral frontoparietal systems[45]. In the context of the dual-network architecture of the multiple demand system, the vFPN has been labeled as sustained task-set maintenance network[34] and has been associated with working memory in the past[14]. Our findings do not support this functional description since tasks strongly implicating working memory (e.g., Encoding or Tower of London) did not dissociate the vFPN from the other FPNs.

One implication of our findings is that FPNs should be functionally defined according to their unique profile of activity across a multivariate task space and, for the fullest functional picture, relative to other functionally similar networks. While it would be possible to try to come up with traditional labels for the underlying cognitive processes in order to understand each of the FPN's unique functional activity profiles (e.g., "linguistic," "response inhibition"), we believe that prematurely labeling brain networks has resulted in much confusion in the field of cognitive neuroscience over recent years and alternative approaches could be more fruitful. Going forward, the unique functional activity profiles across the task space for the different networks that we have uncovered could serve as the building blocks for future work discovering a more accurate neurobiologically-derived cognitive taxonomy.

One possibility is to run further neuroadaptive brain imaging studies but with far greater flexibility in terms of modifiable cognitive tasks along far more dimensions; this may reveal higher-order structures, e.g., in terms of modifiable parameters that more purely relate to network structure, than typical task labels. Alternatively, cognitive tasks could be designed purely behaviorally (e.g., with internet behavioral batteries controlled by Bayesian optimization) such that tasks are maximally unblended and therefore tap into distinct and isolated cognitive processes, which might form a "purer" cognitive task set that will line up more closely with the separate FPNs.

Finally, developments in machine learning mean that single artificial neural networks (ANNs) are becoming able to perform multiple cognitive tests. In a recent study, Yang et al.[46] trained a single ANN on 20 different cognitive tasks. By systematically dissecting the ANN by, for example, "lesioning" clusters of units in this network, the authors could assess how this affected the performance of the ANN on the various tasks. A complete failure of a family of tasks can then provide mechanistic evidence that these tasks share higher-level cognitive processes. Therefore, it may be possible to use ANNs to derive cognitive ontologies that can be projected onto the neural activity profiles derived from neuroadaptive Bayesian optimization. The ultimate goal of using an ANN trained on many cognitive tests would be to provide a generative model of cognitive tasks. The ANN could then be used in conjunction with neuroadaptive Bayesian optimization, to generate highly specialized tasks in real-time that evoke activity uniquely for each of the different FPNs.

Developing detailed functional descriptions of FPNs in terms of cognitive tasks is critical not just for scientific reasons but also for translational applications which require sensitively

discriminating network function, e.g., in developmental or clinical settings. Our results indicate that a major limitation in developing more sensitive diagnostic task batteries to differentiate different types of brain pathology is the need to look at patterns of altered performance across many different tasks, and specific to which underlying networks are being contrasted. Similarly, cognitive training regimes either on their own or in combination with non-invasive brain stimulation are likely to be improved by developing purer cognitive measures that more specifically tap into specific brain networks.

In the present study, neuroadaptive Bayesian optimization allowed an efficient closed-loop search through a large task space, which would have been intractable with standard neuroimaging methodology. Importantly, we found consistent replicable results across and within each individual subject; this is not a trivial advance in light of the growing concerns about the reproducibility of fMRI findings[47–51]. Moreover, the technique transitioned from exploration to a focused search, thereby including an inbuilt replication stage, as it repeatedly kept sampling the predicted optimum by collecting new data (i.e., data previously unseen by prior analysis). This is in line with the recent advocacy of "out-of-sample" prediction in experimental sciences[52,53].

In contrast to meta-analyses, neuroadaptive Bayesian optimization is less biased by prevailing theories of the field, requires far less data and can explore effects that meta-analyses are blind to, such as specific contrasts of interest, functional connectivity or undersampled regions in the task space. Here, we demonstrated that the meta-analysis provides a useful starting point, and that it summarizes many of the network activity profiles well; however, the meta-analysis only provides a partial picture. The discrepancy between the meta-analysis and our results could have arisen for a number of reasons. For example, it could be a consequence of the absence of direct comparisons among different tasks in meta-analyses, differences in baseline or contrast conditions. This needs to be studied in further detail in the future.

Despite its benefits, our results also highlight a number of limitations and areas for future development. In Experiment 3, we found less consistent results in terms of the optimal tasks identified on the subject level, possibly because the BOLD signal averaged across several FPNs yielded smaller target measure values than the single contrast between the dFPN and vFPN, meaning that the contrast-to-noise ratio[54] was lower, and the optimization more challenging[29]. However, when comparing the real-time results from Experiment 3 with the post-hoc results from pooling all data from Experiment 1 and Experiment 3, we find high consistency in the set of optimal tasks identified on the group level, suggesting that our findings are robust. In this respect, it is not clear to what extent the different subject-level optima reflect the choice of acquisition function and the degree to which it is biased towards exploration or exploitation. For example, the acquisition function may have "prematurely" settled on the first optimum identified, whereas an alternative acquisition function could have produced a different outcome: a more explorative acquisition function possibly with a longer optimization period may have resulted in the selection of the same set of cognitive tasks (i.e., mulitple optima) in all subjects.

A related issue is the need to develop robust online stopping criteria for experiments involving real-time optimization. In our experiments, the number of iterations allowed for the optimization was pre-determined before the start of the experiment; however, one avenue for future work is to automatically end the run only when the uncertainty of the algorithm over the experiment space is sufficiently small and/or enough statistical evidence has been accumulated. This can be observed when we look at the subject-level Euclidean distance between successive tasks. For the majority of runs, the scanning time could have been

reduced by several task blocks, for others a longer optimization period could have resulted in more stable results. While we have proposed two online stopping criteria in the past that rely on characteristics of the acquisition function[55], more work is needed to assess how well these perform in more challenging experiment spaces such as those in the present study.

An alternative explanation for different optima between subjects is that this could potentially reflect individual biases towards specific task-network relationships; this could not be assessed in the current work because of the relatively small sample size. Future work could explore the possible existence of inter-individual differences in the profile of activity across tasks and relate these to comprehensive behavioral testing.

Another potential limitation is the parameterization of the 16 tasks in Experiment 1 and Experiment 3. While it is theoretically possible that we discarded tasks as irrelevant due to suboptimal task parameters selected beforehand (e.g., inter-stimulus interval, difficulty level, memory load), we would like to emphasize that this concern is applicable to task-fMRI studies more generally. In the present study, we tried to counteract this problem by selecting "medium" difficulty levels for all tasks. However, in the future we could combine the "coarse" (Experiment 1) and "fine-grained" (Experiment 2) search in a single experiment by spanning a high-dimensional search space consisting of many different cognitive tasks that are simultaneously modifiable along different task parameter dimensions. Another possibility would be to run large behavioral studies testing many different task parameters of cognitive tasks and then select the combination of task parameters that results in similar behavioral performance across all tasks.

In conclusion, the results of the present study have demonstrated the powerful synergy between neuroadaptive Bayesian optimization and meta-analyses for research questions within the cognitive neurosciences that historically have been challenging to tackle. Neuroadaptive Bayesian optimization can go beyond the meta-analysis: using it as a starting point for exploration to find a profile of task activity that maximally dissociate different yet functionally similar brain networks. In so doing, we have discovered a set of unique functional activation profiles across tasks for different FPNs that will form the basis for future work elucidating the many-to-many mapping between cognitive processes and neural systems.

## Methods

**Participants**. Thirty one healthy participants (20 female, 28.39 ± 5.63 years, range: 20–41 years, three left-handed) took part in the real-time optimization study; of these, ten took part in Experiment 1, 11 in Experiment 2, and ten in Experiment 3. For Experiment 2, one participant was excluded due to excessive head movement for both runs. Mean frame-wise displacement (FD)[56] of this participant in both runs was 0.22 and 0.23 mm, respectively; both runs were identified as significant outliers using an iterative implementation of the Grubbs test ($\alpha = 0.05$)[57]. Participants gave written informed consent for their participation. The study was approved by the Hammersmith Hospital (London, UK) research ethics committee. All participants had no history of any neurological/psychiatric disorders and had normal or corrected-to-normal vision. Participants were informed about the real-time nature of the fMRI experiment but no information was given on the actual aim of the study or which parameters would be adapted in real-time. The investigator was not blinded due to the complexity of data acquisition and the need to ensure that real-time optimization was functioning.

**Frontoparietal networks**. The target frontoparietal brain networks were defined based on the meta-analysis reported in Yeo et al.[26]. In this study, 12 cognitive components were identified based on 10,449 experimental contrasts covering 83 BrainMap-defined task categories (BrainMap is a database of functional neuroimaging experiments with coordinate-based results). The authors made publicly available the brain maps that contain the probability of components activating different brain voxels, i.e., Pr(voxel | component), as well as the probability for each task recruiting those components, i.e. Pr(component | task). For Experiment 1 and Experiment 2, we focused on two components within the multiple demand system[11]: Component 8, a ventral frontoparietal network (vFPN) and Component 9, a

dorsal frontoparietal network (dFPN). For Experiment 3, we additionally included two other FPNs: Component 5, a left-lateralized FPN including the inferior frontal gyrus, and Component 6, a bilateral FPN distributed across the medial frontal cortex, the superior parietal cortex and frontal eye fields. Thresholded ($z > 1$) and binarized maps of the four components were used as target networks.

**Experiment space.** For Experiment 1, a 2D-task space was designed based on the same meta-analysis that was used to derive the FPN maps[26]. We selected 16 BrainMap-defined task categories that varied in their probability of recruiting the dFPN and vFPN according to this meta-analysis Pr(component | task). Based on this selection, a 2D-task space was designed with each dimension corresponding to the probabilities of these 16 tasks recruiting the dFPN or vFPN, respectively. A variant of each task was implemented by us in Matlab using Psychophysics Toolbox[58,59]. For a brief description of each of the tasks, please refer to Supplementary Methods. In Experiment 2, for the Deductive Reasoning task, a 1D space was designed with $16 \times 1$ difficulty levels while for the Tower of London task a 2D space was created with 8 (number of steps) $\times$ 2 (convolution) parameters.

The Deductive Reasoning task in Experiment 2 was based on an implementation by Hampshire et al.[14]. A $3 \times 3$ grid of cells was displayed on the screen and each cell contained an object. Each object was made up of three different features: color, shape and number of copies. The features were related to each other according to a set of rules. The subject had to deduce the rules that relate the object features and identify the object whose contents did not correspond to those rules. Task difficulty increased along a single dimension ($16 \times 1$) with increased complexity of the rules applied: problems up to 5 were non-relational; problems 6–9 involved conjunctions that could be solved using the "pop-out effect" (i.e., the answer is a unique stimulus amongst non-unique stimuli); problems from 10 onwards were all relational problems, requiring to work out the conjunction logically. For these problems, we parametrically increased the number of parallel mappings, the level to which they overlap, which in turn added a degree of asymmetry to the problems. Deductive Reasoning problems were presented for 30 s followed by a 5 s response interval, in which subjects were presented with two objects (i.e., the correct one and a randomly selected one), between which subjects chose by pressing the right or left button on a button response box.

Equally, the Tower of London task in Experiment 2 was based on an implementation by Hampshire et al.[14]. Numbered beads were positioned on a tree shaped frame. The participant were instructed to reposition the beads mentally so that they were configured in ascending numerical order running from left to right and top to bottom of the tree. Task difficulty increased along two dimensions ($8 \times 2$); the first dimension represented the total number of moves required (from 2 to 9 moves); the second dimension represented the convolution operation (more or less). Convolution was defined as, when for the participant to succeed on the task a correctly placed bead must first be displaced. Tower of London task problems were presented for 30 s followed by a 5 s response interval, in which subjects were presented with two numbers (i.e., the correct number of moves and the correct plus or minus one number of moves), between which subjects chose by pressing the right or left button on a button response box.

**Hypothesized predictions based on meta-analysis.** The hypothesized predictions were based on the probability values of each task recruiting the FPNs according to the meta-analysis (i.e., Pr(component | task)). For Experiment 1 (Fig. 2b - small panel), we computed the difference between the probability values of the dFPN (Comp 09) and vFPN (Comp 08) for each of the 16 tasks, i.e., Pr (Comp 09 | task)—Pr(Comp 08 | task). For Experiment 3 (Fig. 5b,d - small panels), we computed the difference between probability values of the dFPN/vFPN and the mean of the probability values of the other three FPNs, i.e., Pr(Comp 09 | task)—mean[Pr(Comp 08 | task), Pr(Comp 05 | task), Pr(Comp 06 | task)] for one run and Pr(Comp 08 | task)—mean[Pr(Comp 09 | task), Pr(Comp 05 | task), Pr(Comp 06 | task)] for the other run.

**Experimental procedure.** In Experiment 1, subjects underwent two separate real-time optimization runs. Task space and target brain state were identical across the two runs. For one subject, we only conducted one run due to a failure in the projection to the MR screen. Tasks were presented in a block-wise fashion for 35 s followed by 19 s of rest (black background). Preceding each task, participants received a brief instruction (5 s) about the task they would need to perform in the upcoming block followed by a short rest period (3 s). Each run automatically ended after 20 blocks (20.67 min). For two subjects, one out of the two runs stopped after only 13 blocks due to technical failure.

In Experiment 2, subjects again underwent two separate real-time optimization runs; this time optimizing the tasks that were predicted to be optimal in Experiment 1. The target brain state was identical to Experiment 1 (i.e., dFPN > vFPN). One run optimized for the Deductive Reasoning task while the other run optimized for the Tower of London task. The order of runs was counter-balanced across participants. Each run automatically ended after 20 blocks (18 min). Before each run started, we informed the subjects via microphone which of the two tasks they would need to perform in the upcoming run.

In Experiment 3, subjects also underwent two separate real-time optimization runs. The task space and task presentation was identical to Experiment 1; however

this time one run optimized for the dFPN > 3 FPNs while the other optimized for vFPN > 3 FPNs. The order of runs was counter-balanced across participants. Each run automatically ended after 15 blocks (15.67 min).

In all three experiments, each run was initiated randomly (i.e., the first five blocks were selected randomly from across the experiment space). Subjects were trained and familiarized with all tasks outside of the scanner prior to the start of the experiment. Auditory stimuli were presented using sound-attenuating in-ear MR-compatible headphones (Sensimetrics, Model S14, Malden, USA).

**Real-time fMRI.** Images with whole-brain coverage were acquired in real-time by a Siemens Verio 3T scanner using an EPI sequence (T2*-weighted gradient echo, voxel size: $3.00 \times 3.00 \times 3.00$ mm, field of view: $192 \times 192 \times 105$ mm, flip angle: 80°, repetition time (TR)/echo time (TE): 2000/30 ms, 35 interleaved slices). Prior to the online run, a high-resolution gradient-echo T1-weighted structural anatomical volume (voxel size: $1.00 \times 1.00 \times 1.00$ mm, flip angle: 9°, TR/TE: 2300/2.98 ms, 160 ascending slices, inversion time: 900 ms) and one EPI volume were acquired. Online (as well as subsequent offline) pre-processing were carried out with FSL[60]. The first steps occurred offline prior to the real-time fMRI scan. Those comprised brain extraction[61] of the structural image followed by a rigid-body registration of the functional to the downsampled structural image (2 mm) using boundary-based registration[62] and subsequent affine registration to standard brain atlas (MNI)[63,64]. The resulting transformation matrix was used to register the networks of interest from MNI to the functional space of the respective subject. For online runs, incoming EPI images were motion corrected[64] in real-time with the previously obtained functional image acting as reference. In addition, images were spatially smoothed using a 5 mm FWHM Gaussian kernel. For each TR, means of the FPN masks were extracted. The second stage of pre-processing involved removing large signal spikes with a modified Kalman filter[65]. The pre-processed timecourses were written into a separate text file for subsequent analyses. Removal of low-frequency linear drift and movement artifacts was achieved by adding a linear trend predictor and six motion parameters as confound regressors into the general linear model (GLM). The experiment commenced after 10 TRs to allow for T1 equilibration effects.

**Target measure.** After presentation of each task block, we either calculated the difference in brain level activation between the dFPN and vFPN (Experiment 1 and Experiment 2) or the difference between the dFPN/vFPN and the three other FPNs (Experiment 3). For this purpose, we ran incremental GLMs on the pre-processed timecourses of each FPN separately. In our case, incremental GLM refers to the design matrix growing with each new task block, i.e., the number of timepoints as well as the number of regressors increasing with the progression of the real-time experiment. The GLM consisted of task regressors of interest (one regressor for each block), task regressors of no interest (e.g., 5 s instruction period in Experiment 1/Experiment 3 or 5 s response interval in Experiment 2) as well as confound regressors as described above (six motion parameters and linear trend) and an intercept term. Task regressors were modeled by convolving a boxcar kernel with a canonical double-gamma hemodynamic response function (HRF). After each new block, the beta coefficients were re-estimated. For Experiment 1 and Experiment 2, we computed the difference between the estimates of all task regressors of interest (i.e., beta coefficients) for the dFPN and vFPN (i.e., dFPN > vFPN). For Experiment 3, we computed the difference between the estimates of all task regressors of interest (i.e., beta coefficients) for the dFPN or vFPN over the mean activity of three other FPNs (i.e., dFPN > mean(FPNs) or vFPN > mean(FPNs)). The resulting contrast values were then entered into the Bayesian optimization algorithm. An initial burn-in phase of five randomly selected experimental conditions was employed in both studies, i.e., the first GLM was run after the first five blocks after which the closed-loop experiment commenced.

**Bayesian model.** In the first step of Bayesian optimization, all available samples acquired up to that iteration are used to update the algorithm's surrogate model by predicting the brain's response across the entire experiment space (also for unseen points) using Gaussian Process regression[30–32]. Gaussian processes are fully specified by their mean and covariance functions. As a prior, we employed a zero mean function and as covariance function we chose the squared exponential kernel[32]. The squared exponential kernel encodes the basic prior assumption that points close in the task space elicit similar brain responses while points far from each other could exhibit distinct responses:

$$k(\mathbf{x}, \mathbf{y}) = \sigma^2 \exp\left\{-\frac{(\mathbf{x} - \mathbf{y})^2}{2\,\mathbf{l}^2}\right\}$$

where $\mathbf{x}, \mathbf{y} \in \mathbb{R}^2$ correspond to the choice of experimental condition. The hyperparameters $\sigma \in \mathbb{R}$ and $\mathbf{l} \in \mathbb{R}^2$ each determine the variance and length scale of the covariance kernel, respectively. Further, it is assumed that observations are corrupted by white noise, $\sigma_{noise}^2$. These parameters need to be selected prior to running the experiments. We used independent data from four/two subjects for Experiment 1/Experiment 2 to tune these parameters using Type-2 maximum likelihood[32]. For Experiment 3, we used all data from Experiment 1 to tune these hyper-parameters using Type-2 maximum likelihood. This choice of hyper-parameters was then fixed for all subjects' runs.

**Acquisition function**. In the second step of Bayesian optimization, by maximizing a pre-defined acquisition function, a new point within the experiment space is selected to sample from in the next iteration. In our study, we used the expected improvement (EI) acquisition function that automatically trades-off between exploration and exploitation[30] by studying the predictions of the surrogate model as well as the uncertainty of the predictions. Informally, this choice of acquisition can be seen as trying to maximize the expected improvement (i.e., the target measure) over the current best. We define $m(\mathbf{x})$ as the predictive mean for a point $\mathbf{x} \in \mathbb{R}^2$ and std$(\mathbf{x})$ as the predictive standard deviation. The expected improvement is defined as[30]:

$$EI(\mathbf{x}) = (m(\mathbf{x}) - f_{\max})q(z) + \text{std}(\mathbf{x})p(z)$$

where $q()$ and $p()$ are defined as the cumulative and probability density functions for a standard normal distribution respectively and $f_{\max}$ can be either the maximum predicted or maximum observed value of the objective function depending on our assumptions about the noisiness of the observations[30]. Finally, $z$ is defined as:

$$z = \frac{m(\mathbf{x}) - f_{\max}}{\text{var}(\mathbf{x})}$$

At every iteration, the next experimental condition to be observed is selected by maximizing the expected improvement

$$\mathbf{x}_{\text{next}} = \arg\max_{\mathbf{x}}\{EI(\mathbf{x})\}$$

In Experiment 1 and Experiment 3, $f_{\max}$ was set to be the maximum predicted value. In Experiment 2 we observed much more exploitative behavior of the acquisiton function for the first three subjects; therefore, we set $f_{\max}$ for the final seven subjects to be the maxmimum observed value which lead to increased exploration.

**Real-time sampling behavior of optimization algorithm**. The EI acquisition function is characterized by an automatic transition from explorative to exploitative search behavior when optimization is successful. This results in the acquisition function exploring the experiment space in the beginning of the run, when uncertainty is the greatest, by subsequently proposing experimental conditions far away in the experiment space. Once the algorithm's uncertainty about the experiment space decreases as the run progresses, it transitions into an exploitative phase, in which the acquisition function keeps sampling the predicted optimal experimental condition or nearby in the experiment space. In order to assess if the optimization successfully transitioned from exploration to exploitation in our study, we computed the mean ± std Euclidean distance (ED) between successive experimental conditions across all subjects over course of the run; this distance should decrease with the transition from exploration to exploitation, suggesting successful optimization. Results of these analyses are depicted in Supplementary Fig. 1. In addition, we computed the sampling frequency of each experimental condition for the closed-loop period of the experiment (i.e., iteration 5-end); this contrasts with the burn-in period (i.e., iteration 1–5) in the beginning of the experiment when five experimental conditions were selected randomly. The results of these analyses are depicted in Fig. 2a,c,e and Fig. 5a,c and provide insight about the experimental conditions that were predicted to be optimal across the whole group.

**Bayesian predictions across the experiment space**. In order to obtain group-level Bayesian predictions across the whole experiment space, we conducted Gaussian process regression based on all available observations of all subject. Hyper-parameters of the group-level Gaussian processes were automatically retuned using Type-2 maximum likelihood[32]. The result of the analyses are depicted in Figs. 2b,d and 5b,d and Supplementary Fig. 6 and reveals the experimental conditions that are predicted to be optimal across the whole group.

To better account for subject-specific effects in Experiment 2, we conducted linear mixed effect models. Data entering the linear mixed-effect models were subject-level Bayesian predictions across the task parameters (Supplementary Fig.5,7). For each of the ten subjects this yielded in $16 \times 1$ Bayesian predictions for the Deductive Reasoning and $8 \times 2$ Bayesian predictions for the Tower of London task. We constructed linear mixed-effect models either with a linear or quadratic term for linking Bayesian predictions of brain activity with difficulty/number of steps for the Deductive Reasoning/Tower of London task. For the Tower of London task, we added two additional terms: one linking Bayesian predictions of brain activity with convolution and one interaction term (number of steps × convolution). We then conducted likelihood ratio tests to compare the goodness of fit between the two models (linear vs. quadratic model). Results of this analysis are reported in the Results section.

**Post-hoc cluster-based analyses within dFPN and vFPN**. To demonstrate that the optimization algorithm had successfully selected tasks that targeted the entire dFPN in Experiment 1 and Experiment 2, we conducted post-hoc whole-brain back-projections. For this, we compared the spatial similarity of the group-level Bayesian predictions across the task space (Fig. 2b - large panel) with Bayesian predictions calculated from the timecourse of each voxel in the brain, as detailed

below. This analysis was done on offline-preprocessed data in each subject's native space. Data underwent motion correction[64], spatial smoothing (5 mm FWHM), linear de-trending, and scrubbing[56] (FD threshold > 1.5) before GLMs were conducted on each individual voxels' time series. Resulting beta-coefficients from each voxel's time series were then contrasted with beta-coefficients derived from the vFPN timecourse (as we were interested in the voxel-wise dissociations from the vFPN); and these contrast values were fed into the Gaussian process regression (the Bayesian optimization's model) to derive Bayesian predictions across the whole task space for each voxel. Hyper-parameters of the Gaussian process for each voxel were automatically selected using Type-2 maximum likelihood[32]. To clarify, this offline analyses conceptually only differed from the real-time analysis in the way that we are not running GLMs on the extracted mean timecourse of the dFPN and vFPN (see Methods section—Target Measure), but instead repeating this analysis on each voxels' time series. The Bayesian predictions for each voxel were then spatially correlated with the group-level Bayesian predictions, resulting in one similarity value (Pearson $r$) per voxel. Whole-brain similarity maps were subsequently Fisher $z$-transformed and registered to MNI standard space. For Experiment 1, within-subject runs were averaged before further analyses were conducted. Next, dFPN and vFPN maps were thresholded at $z > 2$ in order to obtain separate clusters within each network. This resulted in five clusters for the dFPN (i.e., paracingulate gyrus, right middle frontal gyrus, left middle frontal gyrus, right superior parietal lobule, and left superior parietal lobule) and seven clusters for the vFPN (i.e., anterior cingulate cortex, right thalamus, left thalamus, right frontal pole, left frontal pole, right frontal operculum, and left frontal operculum); clusters with less than 200 voxels were not considered. For each cluster mask, mean $z$-values of each subject ($n = 10$) were subsequently extracted. Group-level inference was carried out on that data using a one-sample two-tailed $t$-test, and permutation testing was used to correct for multiple comparisons (using $t_{\max}$-method[66]; number of possible permutations was $2^{10}$). Results of this analysis are depicted in Fig. 3.

**Post-hoc cluster-based comparison with meta-analysis**. In Experiment 1, we performed statistical inference to compare group-level Bayesian predictions (Fig. 2b - large panel) with the meta-analysis-based hypothesized predictions (Fig. 2b - small panel) for the entire task space. The same processing pipeline was used as described above; however the Bayesian predictions of each voxel were correlated with the hypothesized meta-analytic predictions, again resulting in one similarity value (Pearson $r$) per voxel. These whole-brain similarity maps were subsequently Fisher $z$-transformed and registered to MNI space. For the whole-network analysis, we extracted mean $z$-values within the dFPN mask for both similarity maps (i.e., hypothesized predictions and group-level predictions) in each subject, and performed a paired two-tailed $t$-test (results reported in Results section). For a more focused analysis, we extracted mean $z$-values for all five clusters within the dFPN separately in each subject (same clusters as described above), and performed a paired two-tailed $t$-test on these values, with permutation testing to correct for multiple comparisons (using $t_{\max}$-method[66]; number of possible permutations was $2^{10}$). Results of this analysis are depicted in Fig. 4.

**Data availability**. For Gaussian process regression, we use a Python implementation from: [http://github.com/SheffieldML/GPy]. Python code for different acquisition functions is available from: [http://github.com/romylorenz/AcquisitionFunction]. All relevant data are available from the authors upon reasonable request.

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

## Acknowledgements

This work was supported by the NIHR Imperial BRC and the Leverhulme Trust. R.Lo. is funded by the EPSRC (P70597). I.R.V. is funded by the Wellcome Trust (103045/Z/13/Z).

## Author contributions

R.Lo. and R.Le. intellectually conceived the study. R.Lo., R.Le., A.H., and I.R.V. designed the study. R.Lo. and R.Le. implemented the experiment with R.P.M., leading the implementation of the Bayesian optimization algorithm under guidance of G.M. R.Lo.

and I.R.V. conducted the experiments. R.Lo. analyzed the results in support of R.Le., A. H., and R.P.M. Lastly, R.Lo., R.Le., and A.H. wrote the manuscript while actively discussing with I.RV.

## Additional information

**Competing interests:** The authors declare no competing interests.

