## [Peer Review File(PDF 2957 kb) · Nature Communications]

Reviewers' comments:

Reviewer #1 (Remarks to the Author):

We have only a vague understanding of the cognitive functions associated with each of a number of brain networks – in particular, the frontoparietal networks (FPNs). As the authors note, the problem is that there is a many-to-many mapping between brain regions and cognitive tasks, and sampling only one or two in any one study is insufficient to characterize a network's role in cognition, thereby contributing to myopic theories. Meta-analyses can be valuable in this regard, but – as reviewed in this submission – have various limitations. The authors have taken a different approach, seeking to develop neurally-derived cognitive taxonomies based on a new method that they've termed neuroadaptive Bayesian optimization. This technique is described in detail elsewhere (Lorenz et al., 2016, 2017), although the authors do a fairly good job of summarizing it here, both in the text and in figures.

Here, the authors seek to dissociate dorsal (dFPN) from ventral FPNs (vFPN), as brain regions in these networks are co-active across multiple tasks. They begin by using a prior meta-analysis to predict which of 16 tasks will be more strongly engaged by the dFPN than the vFPN. They then use these predictions in a guided to select the scanner tasks they will give participants, and they collect real-time fMRI data, estimate the results, and continue to present new tasks from there in such a way as to maximize the difference (dFPN > vFPN). Once they have, in this exploration phase, optimized this difference, they move on to the exploitation phase, wherein they fine-tune the parameters of the tasks that are most diagnostic. The results are unexpected based on the prior meta-analysis. They hasten to note that other results using this approach with other networks are close to what one would have expected, and therefore argue that the results are novel and important.

General Comments

Overall, this is an interesting approach and the paper is clearly written. However, the discussion falls short of the expectations set up in the introduction. First, the authors describe this study as differentiating two networks, but they present a single dissociation – that is, they identify the tasks that yield greater activation for dFPN than vFPN, rather than looking for a double dissociation, which would be more informative. Second, the authors provide very little discussion of what these results tell us about the plausible functions of these two FPN networks. Third, what little they do say focuses on relational integration, because they remark in passing that the higher levels of Deductive Reasoning require more relational integration. This sounds like a possible case of a “misleadingly narrow function being proposed as the definitive role of a network”, to quote the introduction.

Specific Comments

1. It is my assumption, although this is not explicitly stated, that what is plotted on the x and y axes for Stage 1 in Figure 1 is the probability of detecting activation in one of the

networks given a specific task. If this is indeed the case, Tower of London and Deductive Reasoning are two of the tasks that would have the greatest probability of eliciting activation in both dFPN and vFPN. As such, one would expect that these tasks – which turned out to be the most diagnostic – would do a poor job of discriminating between the two networks. By contrast, the ones that should maximally discriminate them would be Posner (vFPN > dFPN) and WCST (dFPN > vFPN) – again, I deduce this from Figure 1, but it is not explicitly stated.

What could account for this discrepancy? Is it simply that Deductive Reasoning and Tower of London are the tasks that engage the dFPN most strongly (and therefore show the biggest difference between dFPN and vFPN)? These tasks require greater deliberation than most/all of the others, and RTs are probably much longer. Similarly, is this the reason that the most diagnostic Deductive Reasoning trials are those from the highest levels of difficulty (albeit not the very highest ones, perhaps because participants tended to guess on those trials)? Does this study tell us anything other than the fact that the most difficult trials of the most difficult tasks are the ones that elicit the greatest activation in dFPN? It may well, but as yet I don't see how.

2. The authors characterize the results as showing that both Deductive Reasoning and Tower of London are diagnostic tasks, but the results are not as compelling for Tower of London. This point doesn't come up in the Discussion.

3. The figure legends do not always provide sufficient information.

Reviewer #2 (Remarks to the Author):

This paper describes the results of a real-time Bayesian optimization approach, which is applied to the problem of disentangling the cognitive processes that activate the dorsal fronto-parietal network (FPN) from those that activate the ventral FPN. The approach is very innovative and shows that a large parameter space can be explored effectively with a small number of subjects, if the right techniques are used. The implications for our understanding of the cognitive processes supported by these regions is interesting, although insufficiently explored. Nevertheless, I think this paper makes a very important contribution to the field. I have a few suggestions for clarification/improvement.

1. The introduction and abstract describe the problem of resolving the functional role of different FPNs. Yet the analyses and results only focus on clarifying the role of the dorsal FPN with respect to the ventral FPN. Therefore, the actual scope of the paper is narrower than the title, abstract and introduction suggest. I think the real scope of the study should be clarified at an earlier point in the manuscript.

2. Given the focus of the paper on dissociating FPNs, the discussion about the interpretation of the results could be more elaborate. It would be interesting to consider what some of the marked differences are between the selected tasks (Deductive reasoning and Tower of

London) and the other tasks that were not able to distinguish ventral and dorsal FPN.

3. In addition, the interpretation of the findings could be discussed more elaborately. What does it mean if a task specifically dissociates dorsal from ventral FPN but does not dissociate between component 5 or 6 and the dorsal FPN? Does it mean that component 5 and 6 are also involved in some processes related to deductive reasoning and planning? And/or is the optimization approach selecting tasks that induce negative activations in the ventral FPN due to the contrast that is used? In relation to this, it would also help to show the BOLD activation for the dorsal and ventral FPN separately, instead of only plotting the differences.

4. In relation to point 2, I think there should be more information about the nature of the other 14 tasks included in the study. A description similar to that of the Tower of London and the Deductive reasoning tasks should be added to the supplementary materials.

5. For the Deductive reasoning task, it is not clear how the rule complexity was increased.

6. I was wondering how the authors decided on the number of iterations that were necessary to effectively optimize the search. In supplementary figure 1a, it seems like the optimization was still ongoing at the final iteration (e.g. it did not plateau). Could longer runs have resulted in a significant change in the outcome? It would also be interesting to see supplementary figure 1a for each subject separately. Do some subjects show a much more rapid optimization than others?

7. There is a typo in the explanation of the deductive reasoning task: number of copied should be number of copies.

Reviewer #3 (Remarks to the Author):

In this paper, the authors apply the technique of neuroadaptive Bayesian optimization to the problem of finding the task (and task setting) that most effectively dissociates between the dorsal and ventral FPN. I think this a quite interesting study, well written and well motivated. I also found very interesting the previous authors' manuscripts about the topic (Lorenz et al, 2016, 2017). My main concern is, considering the high bar of the journal, the extent to which the study is incremental with regard to the previous papers - given that the method was introduced in the first paper, and the second paper already provided a thorough description of the caveats and limitations that this sort of approach can tackle. I think the dissociation and interpretation of the different large-scale frontoparietal systems is very relevant, but the paper falls somewhat short of insights about the significance of the results. What are the biological consequences of the task dissociation? what does it mean from a neuroscience point of view? does any conceptual insight about these networks follow from the fact that the method selected these two tasks?

Some other comments:

- The use GP in Stage 2 is slightly unclear to me. Typically, the distance between points in the input space follows naturally from the topology of the space. Here, how is (x-y) defined? I presume this is pre-specified somehow and uses the results of the meta-analysis. Considering the authors' critical comments about meta-analyses in general, I am wondering about the effects of a misspecification of these distances. Do the authors have any comment on what it would happen if the 2D space were established non-accurately?

- Considering the spaces in Stage 2 and 3 are relatively small, have the authors consider merging Stage 2 and 3 into a single search procedure? Put differently, could it be possible that some tasks were regarded as irrelevant just because they were suboptimally parametrised?

- I am wondering why the imagined movement task produces a higher activation in the dFPN than in the motor system (Comp 02). Do the authors think there is a genuine neurophysiological reason, or could it be due to some methodological issue?

- I found the section "Post-hoc cluster-based analyses within dFPN and vFPN" a bit difficult to follow. For example, in "we compared the spatial similarity of the group-level Bayesian predictions across the task space (Fig. 2b-large panel) with Bayesian predictions calculated from the timecourse of each voxel in the brain", what do the authors mean by the "predictions calculated from the timecourse of each voxel"?

Minor comment:

- In page 4 second paragraph, it is a bit confusing to use the word "stage". This "first stage" does not refer to Stage 1, presumably. Please consider to change the word stage from something else.

REVIEWER COMMENTS:

Reviewer #1:

General Comments

Overall, this is an interesting approach and the paper is clearly written. However, the discussion falls short of the expectations set up in the introduction. First, the authors describe this study as differentiating two networks, but they present a single dissociation – that is, they identify the tasks that yield greater activation for dFPN than vFPN, rather than looking for a double dissociation, which would be more informative.

We agree with the reviewer. In the revised manuscript, we now report results of an additional real-time optimization study we conducted in response to the reviewers' comments. In this extra study, besides optimizing for the contrast dorsal FPN > three other FPNs, we now also optimize for the contrast ventral FPN > three other FPNs. This is now explained in detail in the revised manuscript and in our response to the editorial comments.

Second, the authors provide very little discussion of what these results tell us about the plausible functions of these two FPN networks.

In the revised manuscript, we have completely rewritten our Discussion section and elaborate on the interpretation of our results taking into account the new results as well. Please also see our response to the editorial comments above.

Third, what little they do say focuses on relational integration, because they remark in passing that the higher levels of Deductive Reasoning require more relational integration. This sounds like a possible case of a “misleadingly narrow function being proposed as the definitive role of a network”, to quote the introduction.

The structure of the manuscript, incorporating and motivating the new experiment has changed how we present and discuss the results. The revised manuscript considers in more detail functions other than relational integration and explicitly moves away from ascribing narrow functional roles to each network.

Specific Comments:

- 1. It is my assumption, although this is not explicitly stated, that what is plotted on the x and y axes for Stage 1 in Figure 1 is the probability of detecting activation in one of the networks given a specific task. If this is indeed the case, Tower of London and Deductive Reasoning are two of the tasks that would have the greatest probability of eliciting activation in both dFPN and vFPN. As such, one would expect that these tasks – which turned out to be the most diagnostic – would do a poor job of discriminating between the two networks. By contrast, the ones that should maximally discriminate them would be Posner (vFPN > dFPN) and WCST (dFPN > vFPN) – again, I deduce this from Figure 1, but it is not explicitly stated.*

We apologize for not being specific in the previous version of the manuscript and thank the reviewer for this comment. We now include a short paragraph in the Results section explicitly stating our prior hypothesis based on the meta-analysis (p. 4):

“Prior to the real-time experimentation, a 2D-task space was designed with each dimension corresponding to the probability of 16 different cognitive tasks recruiting the dFPN and vFPN (Fig. 1) according to a previous meta-analysis (Yeo et al., 2014). Based on this meta-analysis, we predicted the Wisconsin Card Sorting and Counting/Calculation task to be optimal for maximally dissociating the dFPN from the vFPN (red color) while we would expect the Posner, Anti-Saccade, and Go/No-Go tasks to be best for maximally discriminating the vFPN from the dFPN (blue color).”

What could account for this discrepancy? Is it simply that Deductive Reasoning and Tower of London are the tasks that engage the dFPN most strongly (and therefore show the biggest difference between dFPN and vFPN)? These tasks require greater deliberation than most/all of the others, and RTs are probably much longer. Similarly, is this the reason that the most diagnostic Deductive Reasoning trials are those from the highest levels of difficulty (albeit not the very highest ones, perhaps because participants tended to guess on those trials)? Does this study tell us anything other than the fact that the most difficult trials of the most difficult tasks are the ones that elicit the greatest activation in dFPN? It may well, but as yet I don't see how.

Our results are unlikely to be driven by longer reaction times for the Deductive Reasoning and Tower of London task as all but the Wisconsin Card Sorting task were fixed-paced tasks, that did not rely on a button press by the subject in order to present the next trial. If a subject did not respond, a new trial would be presented to the subject nonetheless. We now include a description of each task in the Supplementary Material.

With regard to the reviewer's comment of task difficulty, we acknowledge that this could be a potential explanation and now include this in the Discussion of our results (p.12-second paragraph):

“One possibility is that the maximum dissociation between dFPN and vFPN occurred for the two cognitive tasks that in general are more difficult than any of the other 14 tasks. Indeed, we observe an increase in the dissociation for both tasks as they become more challenging (in terms of planning steps or relational integration) before the dissociation asymptotes with increasing difficulty. However, there are other tasks that are cognitively challenging but that do not show the same dissociation or show even the reverse pattern, i.e., greater activity for the vFPN than the dFPN (e.g., Divided Auditory Attention or Encoding tasks). Similarly, if the dissociation related just to difficulty then the vFPN>dFPN contrast would be expected to be greatest for very passive tasks such as Fixation Cross. Therefore, while general task difficulty could play a part in the explanation for the dissociation for those tasks, it is unlikely to be the full explanation, and is more likely to reflect difficulty related to specific cognitive processes.”

2. *The authors characterize the results as showing that both Deductive Reasoning and Tower of London are diagnostic tasks, but the results are not as compelling for Tower of London. This point doesn't come up in the Discussion.*

We agree with the reviewer that this point has not been addressed in the previous manuscript. We now include a short paragraph discussing this finding (p. 12-first paragraph):

“While we found highly consistent subject-level results for the Deductive Reasoning task, we observed less consistency across subjects for the Tower of London task. One potential explanation could be the increased complexity of the experiment space spanning two instead of one task dimension. An additional possibility could be that the effect of convolution is relatively subtle compared to the number of steps.”

3. *The figure legends do not always provide sufficient information.*

We have now revised the figure legends to provide more information.

Reviewer #2:

1. *The introduction and abstract describe the problem of resolving the functional role of different FPNs. Yet the analyses and results only focus on clarifying the role of the dorsal FPN with respect to the ventral FPN. Therefore, the actual scope of the paper is narrower than the title, abstract and introduction suggest. I think the real scope of the study should be clarified at an earlier point in the manuscript.*

Instead of only textual revision as suggested by the reviewer, we have substantially broadened the scope of the manuscript by including the results of a new study we conducted that includes two additional FPNs and also optimized to maximise activity of the ventral FPN.

2. *Given the focus of the paper on dissociating FPNs, the discussion about the interpretation of the results could be more elaborate. It would be interesting to consider what some of the marked differences are between the selected tasks (Deductive reasoning and Tower of London) and the other tasks that were not able to distinguish ventral and dorsal FPN.*

In the revised manuscript, we have completely revised our Discussion section and elaborate on the interpretation of our results. In addition, please refer to our response to the editorial comments above.

3. *In addition, the interpretation of the findings could be discussed more elaborately. What does it mean if a task specifically dissociates dorsal from ventral FPN but does not dissociate between component 5 or 6 and the dorsal FPN? Does it mean that component 5 and 6 are also involved in some processes related to deductive reasoning and planning?*

This is an important point raised by the reviewer, which motivated us to consider these two components (i.e., Component 05 and Component 06) in our additional study that specifically aimed at identifying the unique pattern of activity across tasks for the dorsal and ventral FPN relative to all other frontoparietal networks derived from the meta-analysis.

4. *And/or is the optimization approach selecting tasks that induce negative activations in the ventral FPN due to the contrast that is used? In relation to this, it would also help to show the BOLD activation for the dorsal and ventral FPN separately, instead of only plotting the differences.*

We have now added Supplementary Figure 4, which shows BOLD activation for the dFPN and vFPN separately. We found that the selected tasks (i.e., Deductive Reasoning and Tower of London tasks) were not selected because they induced negative activation in the vFPN.

We have included the following text in the revised manuscript (p.6-third paragraph):

“Equally, when estimating a group-level Bayesian model (i.e., GP regression) based on all observations, the Tower of London and Deductive Reasoning tasks were predicted to be optimal for dissociating the two networks (Fig. 2b). This result was highly consistent within and across subjects (Supplementary Fig. 3) and could not be explained by the particular spatial arrangement of the tasks (Supplementary Results) or a negative induced BOLD activation in the vFPN for these tasks (Supplementary Fig. 4).”

5. *In relation to point 2, I think there should be more information about the nature of the other 14 tasks included in the study. A description similar to that of the Tower of London and the Deductive reasoning tasks should be added to the supplementary materials.*

A brief description of the other 14 tasks included in the study, has been added to the Supplementary Material.

6. *For the Deductive reasoning task, it is not clear how the rule complexity was increased.*

We now include additional details in the task description of the Deductive Reasoning task detailing how the rule complexity was increased (p.27):

“Task difficulty increased along a single dimension (16x1) with increased complexity of the rules applied: problems up to 5 were non-relational; problems 6-9 involved conjunctions that could be solved using the “pop-out effect” (i.e., the answer is a unique stimulus amongst non-unique stimuli); problems from 10 onwards were all relational problems, requiring to work out the conjunction logically. For these problems, we parametrically increased the number of parallel mappings, the level to which they overlap, which in turn added a degree of asymmetry to the problems.”

7. *I was wondering how the authors decided on the number of iterations that were necessary to effectively optimize the search. In supplementary figure 1a, it seems like the optimization was still ongoing at the final iteration (e.g. it did not plateau). Could longer runs have resulted in a significant change in the outcome? It would also be interesting to see supplementary figure 1a for each subject separately. Do some subjects show a much more rapid optimization than others?*

The number of iterations was pre-determined before the start of the experiment. The choice was based on our previous study (Lorenz et al., 2016) where we searched through a larger experiment space (i.e., 361 different audio-visual stimuli) but optimized for a simpler target brain state (visual > auditory BOLD activation).

We have added Supplementary Fig. 2 illustrating the Euclidean distance in experimental task space between successive tasks over time for each subject separately for Experiment 1, which describes the optimization process and convergence. We observe that for the majority of runs the scanning time could have been reduced by a few task blocks, for others a longer optimization period could have potentially resulted in more stable results.

This shows the importance of future work developing online stopping criteria; this is mentioned now in the Discussion (p.17-18):

“A related issue is the need to develop robust online stopping criteria for experiments involving real-time optimization. In our experiments, the number of iterations allowed for the optimization was pre-determined before the start of the experiment; however, one avenue for future work is to automatically end the run only when the uncertainty of the algorithm over the experiment space is sufficiently small and/or enough statistical evidence has been accumulated. This can be observed when we look at the subject-level Euclidean distance between successive tasks (Supplementary Fig. 2). For the majority of runs, the scanning time could have been reduced by several task blocks, for others a longer optimization period could have resulted in more stable results. While we have proposed two online stopping criteria in the past that rely on characteristics of the acquisition function (Lorenz et al., 2015), more work is needed to assess how well these would perform in more challenging experiment spaces such as those in the present study.”

8. *There is a typo in the explanation of the deductive reasoning task: number of copied should be number of copies.*

This has been corrected.

Reviewer #3:

My main concern is, considering the high bar of the journal, the extent to which the study is incremental with regard to the previous papers - given that the method was introduced in the first paper, and the second paper already provided a thorough description of the caveats and limitations that this sort of approach can tackle. I think the dissociation and interpretation of the different large-scale frontoparietal systems is very relevant, but the paper falls somewhat short of insights about the significance of the results. What are the biological consequences of the task dissociation? what does it mean from a neuroscience point of view? does any conceptual insight about these networks follow from the fact that the method selected these two tasks?

Second, the authors provide very little discussion of what these results tell us about the plausible functions of these two FPN networks.

We agree with the reviewer. In the revised manuscript, we now report results of an additional real-time optimization study we conducted in response to the editor's and reviewers' comments. This new study allowed us to study the *unique* functional profile of the dFPN and vFPN. Furthermore, we substantially revised and expanded the discussion of our findings with regard to their neurobiological role. Please also see our response to the editorial comments above.

Some other comments:

- 1. The use GP in Stage 2 is slightly unclear to me. Typically, the distance between points in the input space follows naturally from the topology of the space. Here, how is (x-y) defined? I presume this is pre-specified somehow and uses the results of the meta-analysis. Considering the authors' critical comments about meta-analyses in general, I am wondering about the effects of a misspecification of these distances. Do the authors have any comment on what it would happen if the 2D space were established non-accurately?*

To clarify the design of the 2D task space, further explanation is now provided in legend of Fig. 1 (p.20). In addition, we have added the following paragraph to the Methods section of the manuscript (p.27):

"For Experiment 1, a 2D-task space was designed based on the same meta-analysis that was used to derive the frontoparietal network maps (Yeo et al., 2014). We selected 16 *BrainMap*-defined task categories that varied in their probability of recruiting the dFPN and vFPN according to this meta-analysis $Pr(\text{component} | \text{task})$. Based on this selection, a 2D-task space was designed with each dimension corresponding to the probabilities of these 16 tasks recruiting the dFPN or vFPN, respectively."

Gaussian processes (GPs) are well suited for our optimization problem because they can be applied to complex objective functions: e.g., that are non-convex, feature multiple optima or involve categorical inputs as is the case for the tasks space described above.

Our results indicate that neuroadaptive Bayesian optimization based on GPs seems to deal well even when searching across a task space that possibly contains certain misspecification of distances among tasks. This is evident in the fact that we found highly replicable results that were not predicted by the meta-analysis (Experiment 1), indicating that we were exposed to some degree of misspecification in the tasks space. Further, we were able to identify multiple optima that were not close to each other in the task space (Experiment 3 – especially for the vFPN), demonstrating the benefits of the approach for global and multimodal optimization (i.e., multiple optima), overcoming misspecifications in the tasks space.

In general and throughout our results, we also observe some degree of smoothness in the task space, indicating that tasks eliciting a similar brain response are indeed grouped closer together than tasks with different brain responses. To illustrate that this finding is not just encoded in the GP kernel, we also plot the mean value for each cell for most of our results (see Supplementary Material).

2. *Considering the spaces in Stage 2 and 3 are relatively small, have the authors consider merging Stage 2 and 3 into a single search procedure? Put differently, could it be possible that some tasks were regarded as irrelevant just because they were suboptimally parametrised?*

This is an excellent point by the reviewer. This will be the focus of future work, which will allow us to explore and overcome technical challenges of merging the approaches of Experiment 1 and Experiment 2 in a single procedure (Stage 2 and Stage 3 in the previous manuscript). It is plausible that if the task space becomes too high-dimensional (let's assume 16 tasks varying on e.g. two different task parameter dimensions), that the optimization takes too long and would not reliably identify optima within an acceptable amount of time (~ maximum of 20-30 min for each run).

The other point the reviewer raised about the initial parameterisation of tasks and if some tasks were ignored by the optimization algorithm because they were sub-optimally parameterised is valid and important. This is a plausible concern for any fMRI study – not just the ones that involve real-time optimization. In our present study, we tried to overcome this by choosing 'medium' difficulty levels for all tasks. We now include a paragraph in the discussion section discussing this concern and proposing more systematic ways of tackling this important problem in the future (p. 18):

“Another potential limitation is the parameterization of the 16 tasks in Experiment 1 and Experiment 3. While it is theoretically possible that we discarded tasks as irrelevant due to suboptimal task parameters selected (e.g., inter-stimulus interval, difficulty level, memory load), we would like to emphasize that this concern is also applicable to task-fMRI studies more generally. In the present study, we tried to counteract this problem by selecting “medium” difficulty levels for all tasks. However, in the future we could combine the “coarse” (Experiment 1) and “fine-grained” (Experiment 2) search in a single experiment by spanning a high-dimensional search space consisting of many different cognitive tasks that are modifiable along different task parameter dimensions. While Bayesian optimization is particularly well suited for efficiently searching through high-dimensional spaces (Brochu et al., 2010; Shahriari et al., 2016) it is still an outstanding question how well this translates to brain imaging studies involving human subjects. Another possibility would be to run large behavioral studies testing many different task parameters of cognitive tasks and then selecting the combination of task parameters that results in similar behavioral performance across all tasks.”

3. *I am wondering why the imagined movement task produces a higher activation in the dFPN than in the motor system (Comp 02). Do the authors think there is a genuine neurophysiological reason, or could it be due to some methodological issue?*

Because we have changed the focus of the manuscript more explicitly onto four different FPNs, this figure has been removed.

4. *I found the section "Post-hoc cluster-based analyses within dFPN and vFPN" a bit difficult to follow. For example, in "we compared the spatial similarity of the group-level Bayesian predictions across the task space (Fig. 2b-large panel) with Bayesian predictions calculated from the timecourse of each voxel in the brain", what do the authors mean by the "predictions calculated from the timecourse of each voxel"?*

Thanks for this feedback. Below, we briefly summarize what we did in the real-time experiment.

For Experiment 1 and Experiment 2, we extracted the mean timecourse of the dFPN and vFPN. On these two extracted time courses, we ran separate GLMs. We then computed the difference between the estimates (i.e., beta coefficients) of all task regressors of interest for the dFPN and vFPN (i.e. dFPN > vFPN). The resulting contrast values were then entered into the Bayesian optimization algorithm that used these values to predict the subjects' brain response across the whole tasks space using GP regression (i.e., also for “unseen” tasks). This is explained in detail in the Methods section “Neuroadaptive Bayesian optimisation – Target Measure”.

Now for the post-hoc analyses, we repeated this analysis, but instead of running GLMs on the mean timecourse of the dFPN and vFPN, we ran it on each individual voxels' time series (following some offline pre-processing the data). But the rest of the procedure was identical. The resulting contrast values were then entered into the Bayesian optimization algorithm that used these values to predict the subjects' brain response across the whole tasks space using GP regression. This is what we termed the "*predictions calculated from the timecourse of each voxel*".

To make this clearer in the revised manuscript, we now include additional clarification in the Methods section – "Post-hoc cluster-based analyses within dFPN and vFPN" (p.34):

"To clarify, this offline analyses conceptually only differed from the real-time analysis in the way that we are not running GLMs on the extracted mean timecourse of the dFPN and vFPN (see Methods - Neuroadaptive Bayesian optimization: Target Measure), but instead repeating this analysis on each voxels' time series."

Minor comment:

5. *In page 4 second paragraph, it is a bit confusing to use the word "stage". This "first stage" does not refer to Stage 1, presumably. Please consider to change the word stage from something else.*

Thanks for pointing this out. Instead of referring to Stage 1 in the first paragraph, we now refer to Experiment 1. For this reason, we left the description "first stage" in the second paragraph, as now it should be clear that this refers to the Neuroadaptive Bayesian optimization procedure.

Reviewers' comments:

Reviewer #1 (Remarks to the Author):

The authors have been responsive to the reviews, and have even run a third experiment to address the issue of a single-dissociation between the two key fronto-parietal networks. Unfortunately, the results of the third experiment have not succeeded in clarifying the unique cognitive functions of any of the FPNs. Additionally, the results of the third experiment are more consistent with the Yeo et al. metaanalysis than the previous ones were, and therefore the claim that this approach yields completely discordant results from meta-analysis is no longer quite as strong. This is a methodologically interesting study, and the approach could be used fruitfully to further explore functional dissociations between networks, but this particular study does not provide much theoretical insight relative to prior cognitive neuroscience studies. This manuscript may be better suited to Nature Methods, or another methods-focused journal.

Reviewer #2 (Remarks to the Author):

In this revised version, the authors have added an additional experiment to the paper. This addition has resulted in more interpretable functional profiles of different FPNs and a better alignment between the introduction, results and discussion. There are still a number of things that could be clarified a bit better/more precisely in the paper.

- Comparisons of the obtained results with predictions based on the meta-analysis results play an important role in the paper. However, it is still not completely clear how these predictions were made and how the tasks were selected. Specifically, it is unclear if the authors used the contrast of interest (dFPN>vFPN) to identify the tasks and predictions? Or was it based on probability values of specific tasks activating either one of these networks (activity in vFPN and activity in dFPN separately)? After relevant task categories were selected, was one task exemplar randomly selected from each of these task categories?
- What were the components labels for the vFPN and dFPN in the Yeo et al. (2014) paper?
- The results in experiment 3 seem to be very variable across participants. Therefore it could be that the mean results reported in the paper were actually only driven by a small subset of participants. In order to give the reader some idea of the generalizability of these findings over participants, would it be possible to give some indication of how well the group mean effects (e.g. group mean BOLD differences) distinguish the networks of interest in each participant (vFPN>other FPN and dFPN>other FPN)?
- The label of the colorbar in figure S8 is incorrect

- In the introduction, the authors state: "We identify and subsequently refine two cognitive tasks (Deductive Reasoning and Tower of London) that maximally dissociate these two FPNs." It should already be clarified at this point that this statement refers to the dissociation of dFPN>vFPN.

Reviewer #3 (Remarks to the Author):

The authors have addressed my concerns satisfactorily. Most importantly, they have broadened the interpretation of the results and framed them better within previous literature.

Point-to-point response to reviewers' comments

We thank the reviewers for their supportive and valuable comments, and have revised the manuscript accordingly. Please find our detailed response below; changes are also highlighted in blue in the revised manuscript.

Reviewers' comments:

Reviewer #1 (Remarks to the Author):

The authors have been responsive to the reviews, and have even run a third experiment to address the issue of a single-dissociation between the two key fronto-parietal networks. Unfortunately, the results of the third experiment have not succeeded in clarifying the unique cognitive functions of any of the FPNs. Additionally, the results of the third experiment are more consistent with the Yeo et al. metaanalysis than the previous ones were, and therefore the claim that this approach yields completely discordant results from meta-analysis is no longer quite as strong. This is a methodologically interesting study, and the approach could be used fruitfully to further explore functional dissociations between networks, but this particular study does not provide much theoretical insight relative to prior cognitive neuroscience studies. This manuscript may be better suited to Nature Methods, or another methods-focused journal.

We thank the reviewer for agreeing with the methodological novelty of our study. We would like to emphasise that the aim of our study was never to completely disprove the meta-analysis by Yeo et al. (2014). However, as thoroughly discussed in our manuscript, the results from Experiment 3 still diverge significantly from many of the predictions made by the meta-analysis as well as previous functional descriptions associated with these two frontoparietal networks.

Further, from a theoretical perspective, in the revised manuscript we clearly show that mapping a single function to a single network is not always correct at the macroscopic level (i.e., many-to-many mapping). While admittedly such models produce simple labels for people to use, they are incomplete, and in our manuscript, we detail alternative approaches going forward.

Reviewer #2 (Remarks to the Author):

In this revised version, the authors have added an additional experiment to the paper. This addition has resulted in more interpretable functional profiles of different FPNs and a better alignment between the introduction, results and discussion. There are still a number of things that could be clarified a bit better/more precisely in the paper.

• Comparisons of the obtained results with predictions based on the meta-analysis results play an important role in the paper. However, it is still not completely clear how these predictions were made and how the tasks were selected. Specifically, it is unclear if the authors used the contrast of interest (dFPN>vFPN) to identify the tasks and predictions? Or was it based on probability values of specific tasks activating either one of these networks (activity in vFPN and activity in dFPN separately)? After relevant task categories were selected, was one task exemplar randomly selected from each of these task categories?

The task selection procedure, we have described in detail in the Methods section (p.25-26) of the previous manuscript (section: "Frontoparietal networks" and "Experiment space"):

"The target frontoparietal brain networks were defined based on the meta-analysis reported in Yeo et al. (Yeo et al., 2014). In this study, 12 cognitive components were identified based on 10,449 experimental contrasts covering 83 *BrainMap*-defined task categories (*BrainMap* is a database of functional neuroimaging experiments with coordinate-based results). The authors made publicly available the brain maps that contain the probability of components activating different brain voxels, i.e. $Pr(\text{voxel} | \text{component})$, as well as the probability for each task recruiting those components, i.e. $Pr(\text{component} | \text{task})$. [...]

For Experiment 1, a 2D-task space was designed based on the same meta-analysis that was used to derive the frontoparietal network maps (Yeo et al., 2014). We selected 16 *BrainMap*-defined task categories that varied in their probability of recruiting the dFPN and vFPN according to this meta-analysis $Pr(\text{component} | \text{task})$."

We have added an additional sentence to the revised manuscript to clarify that based on the task categories one possible implementation of this task category was selected by us (p.26):

“Based on this selection, a 2D-task space was designed with each dimension corresponding to the probabilities of these 16 tasks recruiting the dFPN or vFPN, respectively. A variant of each task was implemented by us in Matlab using Psychophysics Toolbox (Brainard, 1997; Pelli, 1997). For a description of each of the tasks, please refer to Supplementary Methods.”

Furthermore, we agree with the reviewer that the predictions based on the meta-analysis results play an important role in our manuscript. These predictions were based on the probability values of each task recruiting each network $Pr(\text{component} | \text{task})$, and then calculating each contrast. To make this more explicit, we have added the following paragraph to the manuscript (p. 27):

“The hypothesized predictions were based on the probability values of each task recruiting the FPNs according to the meta-analysis (i.e., $Pr(\text{component} | \text{task})$). For Experiment 1 (Figure 2b – small panel), we computed the difference between the probability values of the dFPN (Comp09) and vFPN (Comp08) for each of the 16 tasks, i.e., $Pr(\text{Comp09} | \text{task}) - Pr(\text{Comp08} | \text{task})$. For Experiment (3 Figure 5b/d – small panels), we computed the difference between probability values of the dFPN/vFPN and the mean of the probability values of the other three FPNs, i.e., $Pr(\text{Comp09} | \text{task}) - \text{mean}[Pr(\text{Comp08} | \text{task}), Pr(\text{Comp05} | \text{task}), Pr(\text{Comp06} | \text{task})]$ for one run and $Pr(\text{Comp08} | \text{task}) - \text{mean}[Pr(\text{Comp09} | \text{task}), Pr(\text{Comp05} | \text{task}), Pr(\text{Comp06} | \text{task})]$ for the other run.”

- What were the components labels for the vFPN and dFPN in the Yeo et al. (2014) paper?

In the previous version of the manuscript, we mentioned the component labels in the Methods section (p. 25):

“For Experiment 1 and Experiment 2, we focused on two components within the multiple demand system (Duncan & Owen, 2000): Component 8, a ventral frontoparietal network (vFPN) and Component 9, a dorsal frontoparietal network (dFPN).”

However, we understand it would facilitate the reader to directly compare our results with the Yeo et al. (2014) study if we explicitly use the Yeo et al component labels in the main text of the manuscript and in the figures. We have therefore modified the Introduction as follows (p.3):

“To address this question, we went back to the meta-analysis (Yeo et al., 2014) and selected all remaining FPNs (in addition to the dFPN (Comp 09) and vFPN (Comp 08)), resulting in two additional FPNs: a left-lateralized FPN including the inferior frontal gyrus (Fig. 6 – Comp 05) and a bilateral FPN distributed across the medial frontal cortex, the superior parietal cortex and frontal eye fields (Fig. 6 – Comp 06).”

In addition, we added the component labels in Fig. 6:

• The results in experiment 3 seem to be very variable across participants. Therefore it could be that the mean results reported in the paper were actually only driven by a small subset of participants. In order to give the reader some idea of the generalizability of these findings over participants, would it be possible to give some indication of how well the group mean effects (e.g. group mean BOLD differences) distinguish the networks of interest in each participant (vFPN>other FPN and dFPN>other FPN)?

We thank the reviewer for asking us to clarify this. We have performed an additional analysis in which we conducted a spatial correlation between each subject's predictions and the group mean predictions. The results of this analysis have been added to Supplementary Fig. 8 and the analysis is described in the figure legend. Based on these results, we see that individuals' results moderately to highly correlate with the group effects, and that the group results are not driven by a small subset of participants.

We also mention this new analysis in the main text (p.9):

“It should be noted that for both contrasts the acquisition function sampled more broadly than in Experiment 1. This can also be observed when considering individual subjects' results (Supplementary Fig. 8), indicating less consistent optima across subjects. **However, in an additional analysis, we observe that the group-level results are consistent with the individuals' results (spatial correlation coefficients between group and subject-level results in Supplementary Fig. 8).**”

Supplementary Fig. 8:

Description of analysis in Supplementary Fig.8 legend:

“We performed an additional analysis comparing group-level Bayesian predictions (Fig. 5b/5d) with subject-level Bayesian predictions (‘GP’ panels) using Spearman correlation; the results of this analysis are provided here next to the ‘mean’ panels in grey ($r = .XY$). Based on these results, we see that individuals’ results moderately to highly correlate with the group effects, and that the group results are not driven by a small subset of participants.”

- *The label of the colorbar in figure S8 is incorrect*

We thank the reviewer for spotting this. This has now been corrected (see Supplementary Fig.8 above).

- *In the introduction, the authors state: “We identify and subsequently refine two cognitive tasks (Deductive Reasoning and Tower of London) that maximally dissociate these two FPNs.” It should already be clarified at this point that this statement refers to the dissociation of dFPN>vFPN.*

This has been changed now (p. 1):

“We identify and subsequently refine two cognitive tasks (Deductive Reasoning and Tower of London) that maximally dissociate the dorsal from ventral FPN.”

Reviewer #3 (Remarks to the Author):

The authors have addressed my concerns satisfactorily. Most importantly, they have broadened the interpretation of the results and framed them better within previous literature.

REVIEWERS' COMMENTS:

Reviewer #2 (Remarks to the Author):

The authors have addressed all of my remaining concerns.